# Exploring TRF2-Dependent DNA Distortion Through Single-DNA Manipulation Studies

Xiaodan Zhao[1], Vinod Kumar Vogirala[2,3], Meihan Liu[4], Yu Zhou[4], Daniela Rhodes [2,5,6], Sara Sandin [2,5,7✉] & Jie Yan [1,4,8✉]

TRF2 is a component of shelterin, a telomere-specific protein complex that protects the ends of mammalian chromosomes from DNA damage signaling and improper repair. TRF2 functions as a homodimer and its interaction with telomeric DNA has been studied, but its full-length DNA-binding properties are unknown. This study examines TRF2's interaction with single-DNA strands and focuses on the conformation of the TRF2-DNA complex and TRF2's preference for DNA chirality. The results show that TRF2-DNA can switch between extended and compact conformations, indicating multiple DNA-binding modes, and TRF2's binding does not have a strong preference for DNA supercoiling chirality when DNA is under low tension. Instead, TRF2 induces DNA bending under tension. Furthermore, both the N-terminal domain of TRF2 and the Myb domain enhance its affinity for the telomere sequence, highlighting the crucial role of multivalent DNA binding in enhancing its affinity and specificity for telomere sequence. These discoveries offer unique insights into TRF2's interaction with telomeric DNA.

[1] Department of Physics, National University of Singapore, 117551 Singapore, Singapore. [2] School of Biological Sciences, Nanyang Technology University, 637551 Singapore, Singapore. [3] Electron Bio-Imaging Centre (eBIC), Diamond Light Source, Harwell Science and Innovation Campus, Didcot, UK. [4] Mechanobiology Institute, National University of Singapore, 117411 Singapore, Singapore. [5] NTU Institute of Structural Biology, Nanyang Technology University, 636921 Singapore, Singapore. [6] Medical Research Council, Laboratory of Molecular Biology, Cambridge, UK. [7] Umeå university, KBC-huset (KB), Linnaeus väg 10, Umeå 90187, Sweden. [8] Joint School of National University of Singapore and Tianjin University, International Campus of Tianjin University, Binhai New City, Fuzhou 350207, China. ✉email: sara.sandin@umu.se; phyyj@nus.edu.sg

The length and structure of telomeres play a crucial role in maintaining genome stability and cell viability, as well as in the processes of aging and cancer. Telomeres are composed of long arrays of short G-rich sequence motifs, which are capped by six shelterin proteins that form a protective structure that prevents degradation and end-to-end fusions of linear eukaryotic chromosomes. In humans, the double-stranded region of telomeric DNA is about 10,000 base pairs long, followed by a short single-stranded G-overhang. This DNA contains a unique TTAGGG repeat sequence that is evolutionarily conserved[1,2]. The double-stranded region is decorated with two Telomeric-repeat Binding Factors (TRF1 and TRF2), which serve to recruit other shelterin components to telomeres. TRF2 is of particular interest as its depletion leads to telomere deprotection and fusions. Both TRF proteins have a TRFH domain (Telomeric Repeat Factors Homology) and a C-terminal Myb DNA-binding domain (DBD), which are connected by long flexible linkers. The TRFH domain is responsible for homodimerization of TRF monomers and interaction with proteins that regulate telomere maintenance[3,4]. The Myb domain recognizes and binds to the double-stranded region of telomeric DNA. One notable difference in structure between the TRFs is that TRF1 is rich in acidic residues at its N-terminus, while TRF2 is rich in basic residues.

TRF2 is abundant throughout the cell cycle and plays a crucial role in regulating telomere length and function by interacting with multiple telomere-associated proteins. Deletion or inhibition of TRF2 activates the ATM kinase pathway and triggers p53-dependent apoptosis[5,6]. Deletion of TRF2 also leads to frequent NHEJ of telomeres[7,8]. TRF2 is also involved in the formation of the t-loop, a triple-stranded structure formed by the invasion of the G-overhang into the double-stranded region of telomeric DNA[9–11]. This lasso-like structure is thought to protect the telomere end by hiding it from DNA damage repair responses. While we understand that TRF2 binds to telomeric DNA and plays a crucial role in telomere structure and function, many aspects of its interaction with DNA still remain unclear.

The high-resolution X-ray crystallographic analysis of the DNA-binding domains (DBDs) of TRF1 and TRF2 has revealed that DNA binding is mediated by three G-C base pairs and that both proteins interact with DNA in the same manner. The crystal structures showed no evidence of DNA bending or twisting upon binding to the DBDs[3]. Electron microscopy (EM) and atomic force microscopy (AFM) studies on full-length TRF2 have revealed that the protein is capable of forming individual nucleoprotein units[12–15]. A gel electrophoretic mobility assay reported that TRF2 could generate positive supercoiling on circular DNA molecules relaxed by TOPO I, indicating a preference for TRF2 to bind to positively supercoiled DNA[12]. Dual resonance frequency enhanced electrostatic force microscopy (DREEM) imaging of TRFH-bound DNA has demonstrated that TRFH can form individual nucleoprotein units that absorb DNA segments of regular length (~26 nm), which is similar to the DNA length absorbed by full-length TRF2. This suggests that DNA encircles around the TRF2 protein surface in the TRF2 nucleoprotein complex[15]. The model of DNA encircling the protein surface also implies chiral wrapping of DNA, however, there is currently no direct experimental evidence to support this. A deeper understanding of TRF2-DNA structures is crucial in light of the proposed role of TRF2 in mediating DNA deformation to facilitate t-loop formation, as well as to provide further insight into the role of TRF2 in telomere function.

In this study, we aimed to understand TRF2-DNA interactions at a single-molecule level. We used magnetic tweezers[16,17] to conduct single-molecule manipulation experiments. Our findings suggest that TRF2 forms individual nucleoprotein units, and the conformation of DNA bound to TRF2 can vary depending on the salt concentration. Furthermore, we found that DNA bound with TRF2 does not have a strong preference for supercoiling chirality. Our results are consistent with some previous studies but contrast with others. Additionally, we revisited questions about the preferential binding of TRF2 to telomere sequences and the role of individual domains of TRF2 in TRF2-DNA interactions. While these topics have been studied previously[18], our quantification with single-binding site accuracy adds to the previous research in this field.

## Results

**Multiple DNA binding modes of TRF2 are revealed by single-DNA manipulation**. Previously, an Atomic Force Microscopy (AFM) imaging study on full-length TRF2 reported the formation of individual TRF2 nucleoprotein units on telomere DNA, resulting in regular decreases of ~27 nm of DNA per dimer[15]. This study, using dual resonance frequency enhanced electrostatic force microscope (DREEM) for nucleoprotein complexes formed with purified TRFH domain, suggested that DNA could wrap around the TRFH domain, leading to the proposal that TRF2 could form regular DNA wrapping units, similar to nucleosomes, through the mediation of the TRFH domain. However, a later AFM imaging study on full-length TRF2 revealed more diverse conformations of the TRF2-DNA complexes[14], suggesting multiple DNA binding modes for TRF2. As these imaging studies only provided static conformational information of TRF2 in complex with DNA, the stability and dynamics of these TRF2 nucleoprotein complexes remain unclear. To gain insights into these questions, the dynamic DNA deformation upon TRF2 binding to DNA molecules was investigated under different solution conditions using magnetic tweezers.

In the single molecule force manipulation study, DNA deformation by TRF2 binding was probed based on the extension change of DNA. The DNA construct used in these experiments contains a 200 bp DNA-of-interest spanned between two PNA/DNA hybrid handles (Fig. 1a and Supplementary Fig. 1). The PNA/DNA hybrid handles were introduced used to suppress non-specific binding of TRF2 to the handles, which were proven to be highly effective. Within the 200 bp double-stranded (dsDNA), 25 repeats of telomeric DNA sequence 5′-TTAGGG-3′ is sandwiched between a 20-bp and a 30-bp non-specific DNA sequence. The non-B form conformation and the reduced negative charge of the PNA/DNA hybrid handles together ensure that TRF2 has lower propensity to bind to these handles (Supplementary Fig. 1d). Magnetic tweezers record the change of the height of the bead at different forces (refer to Methods). The force-height (F-H) curves of the DNA were measured using several different TRF2 concentrations ranging from 0 to 50 nM. At each concentration, the F-H curve was recorded during a force-decrease scan, followed by a force-increase scan through the same set of forces. At each force, the DNA was held for 20 s to obtain the average extension of the DNA.

Figure 1b shows the force-height (F-H) curves of the DNA at different TRF2 concentrations. The data points from the force-decrease and force-increase scans of naked DNA overlap, demonstrating that the 20-s holding time at each force is sufficient for the DNA conformation to reach equilibrium. With an increase in TRF2 concentration to 20 nM, noticeable DNA shortening compared to naked DNA was observed in the force-decrease scan at forces below 0.3 pN. This shortening was maintained in the subsequent force-increase scan until ~3 pN, resulting in a hysteresis between the F-H curves. This hysteresis indicates that the DNA was shortened at low forces and did not return to its equilibrium state within the time scale of the force-increase scan. This phenomenon was observed in all independent

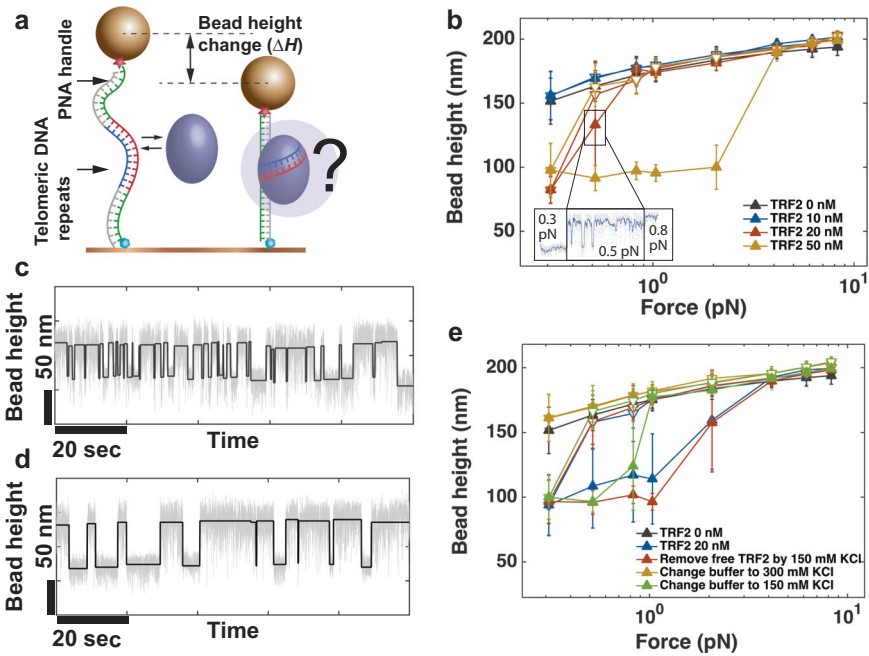

**Fig. 1 Detect TRF2 induced DNA deformation. a** Schematics of probing TRF2 induced dsDNA deformation based on bead height change. **b** Representative force-height curves obtained from a DNA tether in a force-decrease scan (hollow symbols) followed by a force-increase scan (solid symbols) at several different TRF2 concentrations in 150 mM KCl. Inset shows a bead height time trace at ~0.3 pN, ~0.5 pN, and ~0.8 pN revealing the dynamic transition between an extended and a compact TRF2 nucleoprotein conformations at ~0.5 pN. **c, d** Representative fluctuation of a DNA construct recorded at ~0.5 pN in 20 nM TRF2 (**c**) and after removal of free TRF2 (**d**). **e** Representative force-extension curves of a DNA tether recorded at different experimental conditions changed with the top-to-bottom order indicated in the figure panel. The error bars in (**b**) and (**e**) are standard deviations of the bead height during the recording time window.

experiments performed on different DNA molecules (>5) (Supplementary Fig. 2). This result shows that TRF2 can compact the DNA conformation at low levels of tension. The unusually large error bar observed in the F-H data at ~0.5 pN is mainly due to reversible fluctuations between the extended DNA conformation and a highly compact TRF2-induced DNA conformation (as shown in the blue time trace in the inset of Fig. 1b).

In order to determine if the compact DNA conformation observed in the presence of TRF2 is due to the formation of a nucleosome-like unit, we conducted constant force measurements following the aforementioned force-loading scans. The tethers were initially subjected to high forces (>5 pN) to unfold the compact TRF2-DNA complex. Subsequently, the forces were reduced to sub-pN levels to observe the dynamics of TRF2-mediated DNA conformation changes. We searched the forces at which reversible stepwise changes in the bead height could be observed.

Figure 1c illustrates a representative trace of the height of a bead attached to a DNA tether, which was recorded in the presence of 20 nM TRF2 at a force of 0.5 ± 0.1 pN. At this force, fluctuations between different bead heights were observed, with step sizes of around ~50 nm and folding and unfolding rates that were nearly balanced and in the order of 1/s. Similar stepwise fluctuations were observed in seven independent DNA molecules over a force range of (0.3, 0.8) pN. In five of them, the step sizes were distributed around a single peak, while the remaining two showed multiple peaks (Supplementary Fig. 3). Since the DNA (25 telomere repeats and 50 bp non-specific sequence) is capable of accommodating more than one TRF2-DNA nucleoprotein complex, the multi-peak step size profile could be due to the presence of more than one TRF2 on the DNA, which could sterically interfere with each other's compaction activity on the DNA.

To investigate the cause of the stepwise fluctuation in DNA extension, we aimed to determine whether it resulted from TRF2 binding and unbinding or from spontaneous transitions between a compact and extended conformation of the TRF2-DNA complex. After observing the stepwise extension fluctuations, we eliminated the possibility of binding/unbinding fluctuation by removing free TRF2 from the system. To achieve this, we leveraged a well-known observation that DNA-bound proteins often exhibit prolonged lifetimes on DNA when there are no free competing DNA binding proteins[19-22].

After observing the stepwise extension fluctuations caused by TRF2, the free TRF2 was removed from the channel using a 200 μL buffer solution, which was 20 times the channel volume. Despite the removal of free TRF2, stepwise fluctuations were still observed in all experiments (more than five) (Fig. 1d shows a representative time trace). This observation indicates that the TRF2-DNA complex has both a compact and an extended conformation, spontaneously switching between the two conformations at sub-pN forces. DNA compaction at lower forces and de-compaction at higher forces were also observed during force-loading scans (Fig. 1e, the blue curve shows an example). The resulting F-H curves are similar to those recorded in the presence of free TRF2, suggesting that the hysteresis was caused by the force-dependent compaction and decompaction of the bound TRF2-DNA complexes.

Furthermore, we discovered that increasing the salt concentration to 300 mM KCl in the absence of free TRF2 can abolish the TRF2-mediated DNA compaction. However, this is not due to the dissociation of TRF2, since after changing the salt concentration back to 150 mM KCl, stepwise extension fluctuations reappeared consistently in independent experiments (more than five) (Fig. 1e, green curve shows one example). The findings suggest that without competing free proteins present, the nucleoprotein

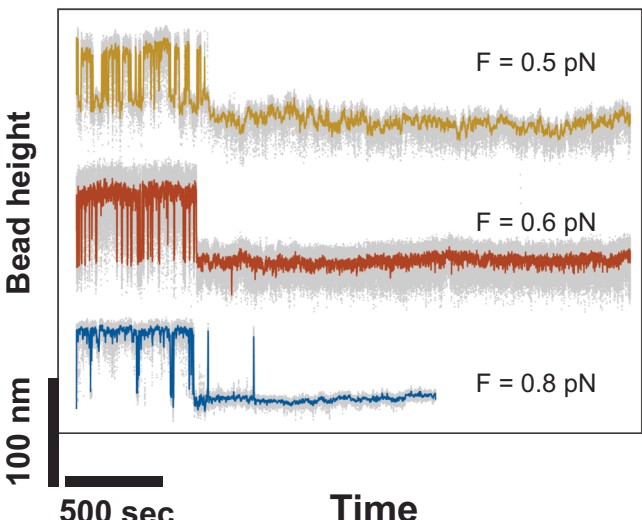

**Fig. 2 Multiple compact nucleoprotein complexes indicated by different stabilities.** Representative time traces of the bead height obtained from three independent tethers after holding the tethers at the corresponding constant forces for long duration. In all cases, after ~500 s, the initial rapid stepwise fluctuation of the bead height eventually transited into a level corresponding a more stable compact nucleoprotein complex.

complex remains stable on the DNA for an extended duration even when the salt concentration is elevated to 300 mM KCl. The observation that increasing the salt concentration can shift the DNA from a compact to an extended conformation without dissociation suggests that the TRF2-DNA complex involves at least two DNA contact points, and that the compaction caused by TRF2 is likely a result of an electrostatic interaction between TRF2 and the DNA.

The results shown in Fig. 2 reveal the behavior of multiple time traces of bead height, recorded from different tethers (more than five) in the force range of 0.3 − 0.8 pN, in the presence of 20 − 50 nM TRF2. In every instance, after holding the tether at the corresponding constant force for a sufficient duration ( > 500 s), the initial rapid stepwise fluctuation of the bead height transformed into a more stable state, where the bead height remained at a lower level. This observation suggests that after the initial binding to DNA, the TRF2 nucleoprotein complex evolves into a more stable conformation, causing DNA compaction. Furthermore, it consistently shows that multiple DNA-compact conformations with varying stability exist.

Three states of DNA conformation are revealed in these time traces, an extended DNA conformation, and two compact DNA conformations but one being more stable than the other. At a sub-pN force, when the more stable compact conformation formed following the less stable, dynamic conformation, it remained in the compact conformation at the same force till the ends of the recording in all experiments we performed. The typical time scale from entering the stable compact conformation till end was hundreds to >1000 s. The (>500 s) time required for the transition from a less stable intermediate compaction conformation to a more stable compact conformation, even at forces as low as ~0.3 pN, highlights the presence of a secondary, slower process of physical organization of the DNA in the nucleoprotein complex, which could be the cause of the hysteresis observed in the force-extension curves recorded at >20 nM TRF2 (Fig. 1b and e). DNA compaction and stepwise fluctuations were also observed in non-telomeric dsDNA of a few kbps (Supplementary Fig. S4), indicating that these observations are not

specific to telomeric sequences or influenced by the limited length of the DNA used to obtain the data in Fig. 2.

The results in this section suggest that complexes formed between TRF2 and DNA sequences on telomeres can adopt either extended or compact conformations, which can be influenced by tension and salt concentration. Furthermore, there appear to be multiple compact conformations with varying levels of stability, the relative stability of which is dependent on the force and salt conditions.

**TRF2 binding does not have strong supercoiling chirality preference.** A previous supercoiling bulk assay using circular DNA plasmids suggests that TRF2 has the ability to generate positive DNA supercoiling. It was proposed that this positive DNA supercoiling generation activity could promote the unwinding of DNA outside the nucleoprotein complex region, leading to the formation of t-loops[12]. It was also suggested that the positive DNA supercoiling generation is due to the right-handed wrapping of DNA on the protein surface[12]. However, the previous bulk assay for supercoiling doesn't provide information on the strength of TRF2 binding in terms of DNA chiral wrapping activity.

To determine if TRF2 binding has a strong preference for DNA supercoiling chirality, we performed a single-DNA supercoiling assay using magnetic tweezers on a non-specific DNA template of 6500 bp in length. We recorded the height of the bead attached to the DNA, which was represented by the $\sigma$-H curve, as a function of supercoiling density $\sigma$ (refer to Methods). The $\sigma$-H curve was measured in the absence of TRF2 (Fig. 3a) and in the presence of different molar concentrations of TRF2 (Fig. 3b–d), at different forces. Figure 3b displays the results obtained at a force of 0.3 ± 0.1 pN, with 0 nM, 5 nM, 10 nM and 15 nM concentrations of TRF2. The measurement process involved sequential changes in $\sigma$ from 0 forward to 0.06 (solid triangles), then backward to 0 (open triangles), forward to -0.06 (solid triangles), and finally backward to 0 (open triangles). The average height of the bead was obtained by holding the DNA at each $\sigma$ value for 10 s. One complete $\sigma$-H curve was recorded in 30 min.

In the absence of TRF2 (0 nM), the profile of the $\sigma$-H curve is symmetrical around $\sigma = 0$. When $|\sigma|$ exceeds ~0.02 (the buckling transition points, indicated by black arrows in Fig. 3a), the DNA winding and unwinding leads to rapid decreases in bead height due to the formation of (+/-) plectonemes[23–26]. In contrast, when TRF2 is present (>5 nM), TRF2 binding results in lower bead height indicating shorter DNA extension. The level of DNA shortening increases with increasing TRF2 concentration, consistent with the formation of compact TRF2-DNA complexes at low tension in the DNA.

At this applied force, despite some hysteresis between the forward and backward curves, the $\sigma$-H curves show nearly symmetrical profiles around $\sigma = 0$, suggesting that TRF2 binding does not cause a substantial shift in the DNA supercoiling center. The same behavior was observed when the experiment was repeated at a higher force of 0.6 ± 0.06 pN (Fig. 3c), with TRF2-induced DNA shortening and symmetrical $\sigma$-H curve profiles. These characteristics remained unchanged when the time for recording the average bead height was increased from 10 s to 60 s, taking 3 h to record one $\sigma$-H curve (Supplementary Fig. 5), indicating a likely near-equilibrium behavior.

While these data do not clearly indicate a pronounced preference of TRF2 for a specific DNA supercoiling chirality in this assay, they do not definitively rule out a stronger right-handed DNA chiral wrapping model of the TRF2 nucleoprotein complex in the absence of tension. Indeed, despite the nearly symmetrical profile of the data obtained at low forces of 0.3 pN

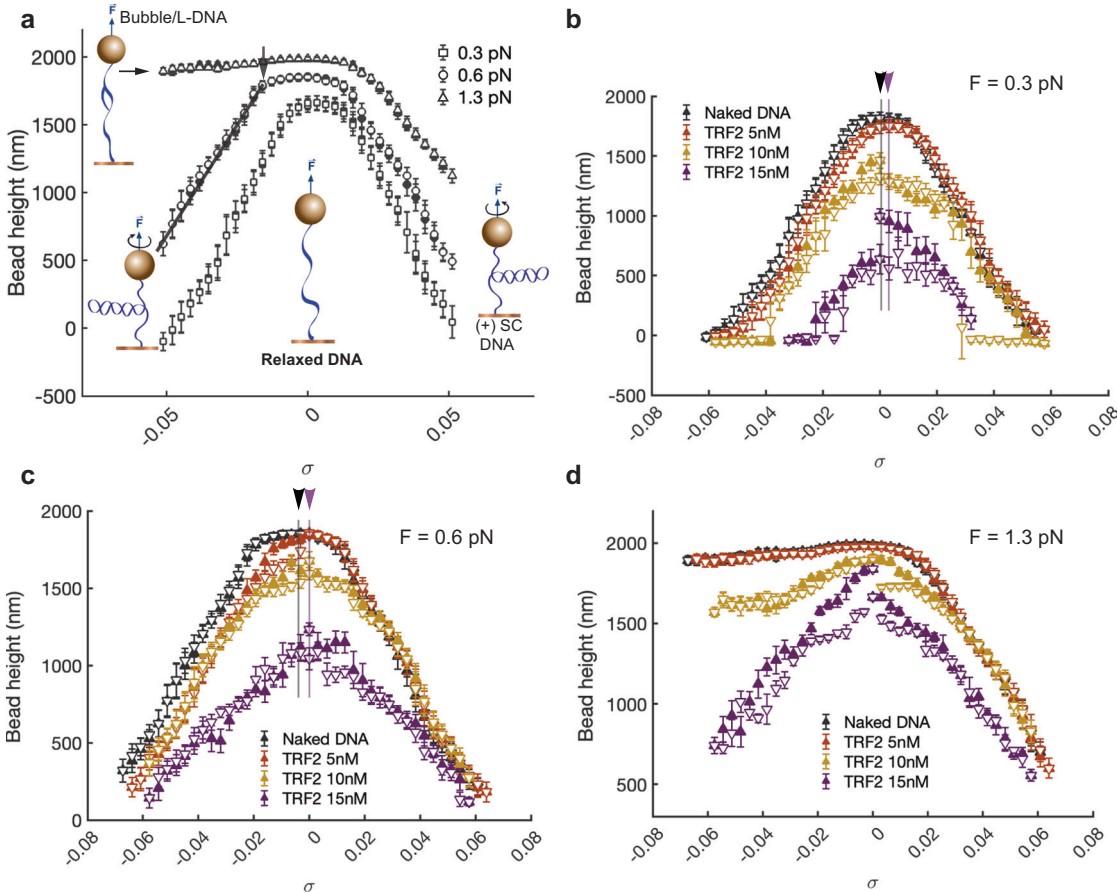

**Fig. 3 Supercoiling assay of TRF2 binding on DNA. a** Representative bead height (H) as a function of DNA superhelical density ($\sigma$) changes at 0.3, 0.6 and 1.3 pN. Insert shows the schematics of different DNA conformations corresponding to tension and superhelical density. The bead height decreases symmetrically as DNA linking number increase and decrease when the force is <0.7 pN, which is due to formation of DNA plectonemes. Unwinding DNA at the force of 1.3 pN, DNA duplex is melted instead of forming DNA plectonemes, therefore, no apparent extension decrease is observed. Representative time traces of bead height as a function of DNA linking number change at varied TRF2 concentrations when 0.3 pN (**b**), 0.6 pN (**c**) and 1.3 pN (**d**) forces were applied. In **b**, **c**, the black and purple downward arrows indicate the apparent supercoiling center in the absence of TRF2 and with the presence of 15 nM TRF2, respectively. The error bars are standard deviations of the bead height during the recording time window.

and 0.6 pN, the supercoiling curves appear slightly skewed toward the positive supercoiling density side at concentrations >10 nM (Fig. 3b, c). This observation may suggest a subtle chiral deformation of DNA under low tension, which could become more pronounced if the force is further reduced.

Additionally, at both forces, the buckling transition threshold points shifted toward $\sigma = 0$ due to TRF2 binding. Since the buckling transition under force is a result from a tug-of-war between the torsion energy and bending energy stored in the DNA, which is also influenced by the applied force[24–26], this result indicates that the TRF2 binding can deform DNA in a manner that can reduce the energy cost of DNA bending for formation of DNA plectonemes. These results were consistently observed in more than five independent experiments (Supplementary Fig. 6). When the experiments were conducted at a higher force of $1.3 \pm 0.1$ pN (Fig. 3d), bare DNA underwent structural transitions to melted or other under-wound non-B DNA structures[25,27–30] during unwinding. However, at a TRF2 concentration of 15 nM, the resulting $\sigma$-H curves remained symmetrical around $\sigma = 0$. This outcome again suggests that TRF2 binding can deform DNA and reduce the energy cost of DNA bending against force, thereby preventing DNA denaturation unwinding at this force, as predicted by theoretical modeling[26].

As DNA wrapping proteins with strong chirality preferences are expected to shift the supercoiling center, as predicted by theory[26] and demonstrated in nucleosome experiments[31] (Supplementary Fig. 7a), these results rule out the possibility of a strong regular DNA wrapping activity with a fixed linking number change. Similarly, DNA wrapping that can flip between two distinct left and right handedness without a preference can also be ruled out, as this would cause the buckling transition threshold to move away from $\sigma = 0$, in contrast to the observed behavior. We also tested whether chiral binding of TRF2 could take a longer time scale to occur, by conducting supercoiling experiments over different time scales up to 600 s. The resulting data still did not reveal a detectable chiral preference of DNA binding by TRF2 (Supplementary Fig. 8). A similar observation was made when using a 2.8 kb PUC-19 plasmid DNA with an insert of 25 repeats of 5'-TTAGGG. This suggests that TRF2 does not show any detectable preference for DNA supercoiling chirality in relation to the telomere sequence at this particular length (Supplementary Fig. 9).

We also investigated the potential preferential chiral binding of TRF2 using an alternative method, which offers higher sensitivity to probe binding-induced chiral changes. This method is based on examining binding-induced alterations in DNA extension to a pre-wound or pre-underwound DNA template[32]. To achieve this,

we introduced a (+) or (-) linking number change at a linking number density of ± 0.03, while maintaining a fixed force of 0.3 pN to facilitate the formation of supercoiled plectonemes. If TRF2 binding exhibits a strong chiral preference, it should relax the supercoiled DNA with the opposite supercoiling chirality, leading to an extension increase. However, after introducing a solution of 20 nM TRF2, we observed further compaction of the DNA induced by TRF2, irrespective of whether it was wound(+) or underwound(-) (Supplementary Fig. 10). Consequently, the results do not provide evidence of detectable chiral binding by TRF2 either. A similar conclusion can be drawn from an alternative experiment where DNA was first incubated with 20 nM TRF2 at 0.3 pN and zero linking number density, followed by unwinding or winding of the DNA (Supplementary Fig. 11).

The data are more consistent with sharp DNA bending that is not associated with substantial chiral deformation, as predicted by theory[26] and demonstrated experimentally by DNA bending proteins such as HMGA2[33]. TRF1, another known DNA bending protein[34], also results in similar changes to the $\sigma$-H curve of DNA (Supplementary Fig. 12).

**The determinants of TRF2-DNA binding affinity.** In previous studies, several factors have been found to influence the binding affinity of TRF2 to DNA, including DNA sequence, the N-terminal basic domain of TRF2, the TRFH domain, and the C-terminal Myb domain[13,35]. In this section, we use a label-free single-molecule quantification assay to quantify the contribution of these factors to TRF2's DNA binding affinity. The assay is based on a short DNA hairpin structure that serves as a protein binding detector[33,36–38]. We determine TRF2's binding affinity using a protocol described in our previous review[39]. In brief, a DNA hairpin held by Watson-Crick base-pairs can mechanically destabilize when force exceeds a threshold $F_c$. The threshold depends on the DNA sequence, solution conditions, and the size of the terminal loop[40]. Reversible transitions between the unzipped and zipped states can only be observed within a narrow force range around $F_c$, $\delta \sim k_B T/\Delta x$, where $k_B$ is the Boltzmann constant, $T$ is the temperature, and $\Delta x$ is the step size of the hairpin unfolding at the critical force $F_c$. The DNA is primarily in the unfolded state at forces above $F_c + \delta$ and in the folded state at forces below $F_c - \delta$.

At low forces ($F_b < F_c - \delta$), the stable DNA hairpin, which is excluded from force transmission, can interact with TRF2 in solution at zero force. When the force is increased to a higher level ($F_p > F_c + \delta$), the hairpin undergoes an instant one-way unfolding transition, resulting in a sudden increase in extension (Fig. 4a, solid black arrow). The height of the bead will continue to increase as the force is further increased from $F_p$ to $F_d$, due to the increased entropic extension of the molecule under the higher force and potential bead rotation caused by the change in force[17] (Fig. 4a, open black arrow). If the hairpin is bound to a protein before the force jumps from $F_b$ to $F_p$, the hairpin will undergo a partial unfolding (Fig. 4a, grey solid arrow 1), pause at the position where the protein is bound, until the protein dissociates, followed by the unfolding of the remaining hairpin (Fig. 4a, grey solid arrow 2). The extended time taken for the hairpin to fully unfold after the force jump serves as a readout of protein binding.

The detection of TRF2 binding to the DNA hairpin is based on the delay in the full unzipping of the hairpin at a binding-probe force ($F_p$). In our experiments, $F_p$ was chosen in such a way that the lifespan of the hairpin at this force was <1 s. Hence, if the delay was >1 s, the DNA was considered to be bound by TRF2 right before the force jump (Fig. 4a, red arrow). To determine the equilibrium probability of TRF2 binding to the DNA hairpin, we repeated a cycle of three forces: a binding force ($F_b$) for a duration of ($\Delta T$) to allow the hairpin to interact with TRF2, a probing

force ($F_p$) as described earlier to determine if the hairpin was bound by TRF2, and a displacement force ($F_d$) that was much >($F_c$) to instantly remove any bound TRF2 and unfold the hairpin. After the TRF2 was removed and the DNA was re-zipped by jumping to ($F_b$), the DNA hairpin was ready for the next round of TRF2 binding. By repeating this cycle for a sufficient number of cycles ($N_{cyc}$), the binding probability was determined by ($p_b^{\Delta T}(c)$), which depended on both the concentration of the protein ($c$) and the duration ($\Delta T$) for binding.

Equilibrium binding can be achieved by employing a sufficiently long duration of $\Delta T$, during which many binding and unbinding events can take place. The resulting equilibrium binding probability is represented by $p_b^{eq}(c)$ and is linked to binding affinity and cooperativity through the equation $p_b^{eq}(c) = \frac{(c/K_d)^n}{1+(c/K_d)^n}$, where $K_d$ is the dissociation constant and $n$ is the Hill coefficient. A Hill coefficient of $n = 1$ signifies non-cooperative binding, while $n > 1$ indicates cooperative binding. TRF2 has a preference for binding at the junction of three-way DNA structures[35], which can impact the assay. To prevent such junctional binding, the hairpin handles were made of PNA/DNA duplexes with low TRF2 binding affinity (Supplementary Fig. 13).

*Selective Binding to Telomere Sequence.* We first examined TRF2 binding to a 52 bp hairpin that consisted of 5 repetitions of 5'-TTAGGG surrounded by a 12 bp left and a 10 bp right non-specific dsDNA region. As previously reported by Court et al.[41], a minimum of 2.5 5'-TTAGGG repeats is required for telomeric-specific binding mediated by the Myb domain. Consequently, a sequence featuring five repeats of 5'-TTAGGG is more than adequate for investigating TRF2's specific binding affinity to the telomeric sequence. Additionally, we discovered that with five repeats of the TTAGGG sequence, the likelihood of G-quadruplex (G4) formation prior to hybridization is minimal, thereby facilitating the experimental process. The non-specific dsDNA regions were added to guarantee full rezipping of the hairpin at $F_b$ (Supplementary Fig. 13).

Representative time traces of the hairpin assay under force-cycle procedures in the absence and presence of 10 nM TRF2 are shown in Fig. 4b. In the presence of 10 nM TRF2, delays in hairpin unzipping were observed (indicated by red arrows in the middle panel). The binding, probing, and displacing forces were set to $F_b = 7.0 \pm 0.7$ pN, $F_p = 13.0 \pm 1.3$ pN, and $F_d = 30 \pm 3.0$ pN, respectively (bottom panel). The DNA hairpin was held at $F_b$ for 20 s, which provides enough time for TRF2 binding and unbinding to the DNA hairpin to reach equilibrium in the absence of force (Supplementary Fig. 14, see Discussion). At $F_p$, the DNA was held for 5 s to measure the hairpin lifetime. The force was then increased to $F_d$ for an additional 5 s to guarantee complete dissociation of any TRF2 that was stably bound to the DNA. For any given DNA tether and TRF2 concentration, an experiment consisted of a minimum of 50 such force-jump cycles.

Figure 4c shows the binding probability as a function of TRF2 concentration for the telomeric DNA hairpin (solid grey circles). The results show that TRF2 binds to the telomeric DNA hairpin with an equilibrium dissociation constant of $K_d = 10.3 \pm 2.9$ nM and Hill coefficient of $n = 1.0 \pm 0.3$ (solid grey circles). Here, the values are the sampled means and errors are sampled standard deviations generated by 1000 resampling of the data points around their data means assuming Gaussian distributions (Supplementary Notes: "Bootstrap analysis"). This value of $K_d$ is consistent with previous measurements on similar lengths of telomere DNA[18]. When the telomere repeats were substituted with non-specific DNA (~52% AT), the results showed a dissociation constant and Hill coefficient of $K_d = 76.8 \pm 31.7$ nM and $n = 1.2 \pm 1.7$, respectively (grey solid triangles). These results demonstrate that TRF2 binds to the telomeric DNA

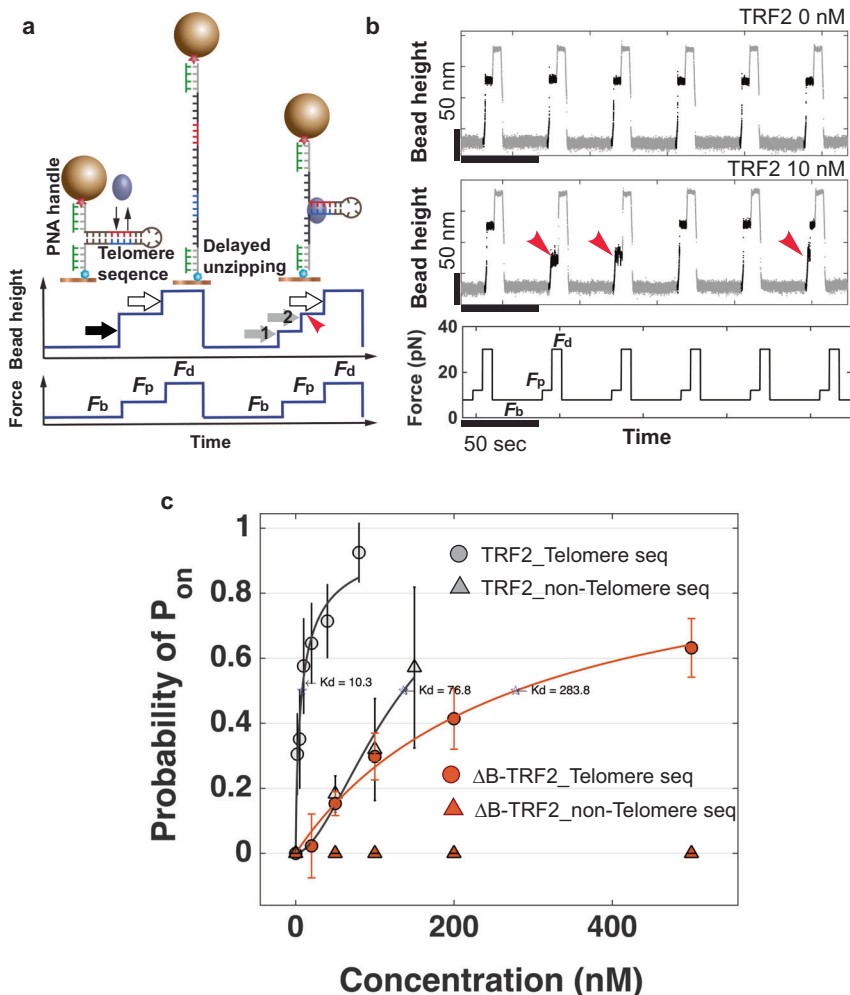

**Fig. 4 Hairpin assay to determine TRF2 binding affinity. a** Schematics of a bead height change of a hairpin containing a specific sequence of interest during two cycles of three forces($F_b$, $F_p$, and $F_d$). The hairpin construct is spanned between two PNA coated 200-nt ssDNA handles. In the first cycle, the hairpin was unzipped immediately after jumping to $F_p$ and comparatively a delayed unzipping is shown in the second cycle. **b** Representative time traces of the height change of a 52-bp hairpin containing 5 repeats of 5'-TTAGGG. During cycles of sequential force jumps among $F_b = 7.0 \pm 0.7$ pN, $F_p = 13.0 \pm 1.3$ pN and $F_d = 30 \pm 3.0$ pN (bottom panel), in the absence (top panel) and presence of 10 nM TRF2 (middle panel, red arrows indicate delayed unzipping). **c** Binding probability as a function of TRF2 (grey) and TRF2$_{45-500}$ (red) concentrations on the DNA hairpin containing the 5 telomeric repeats (circle) and on the nonspecific DNA of the same length (triangle). The error bars represent the standard deviation of independent experimental data.

hairpin with ~7 times higher affinity compared to non-specific DNA of the same length. This selective binding of TRF2 to the telomere sequence suggests that TRF2 preferentially binds to the hairpin stem, instead of the junction, as low-affinity PNA/DNA handles effectively suppress junctional binding.

*The differential contributions of TRF2 domains to TRF2-DNA binding affinity.* We evaluated the impact of various TRF2 domains on the binding affinity of the DNA hairpin containing telomeric sequences, as described earlier. The results, shown in Fig. 4c as solid red circles, depict the binding probability as a function of TRF2$_{45-500}$ concentration (ΔBTRF2). TRF2$_{45-500}$ is a truncated form of TRF2 with the deletion of the basic N-terminal residues 1-44, as detailed in the "Protein purification" section of the Methods. The dissociation constant was calculated to be $K_d = 283.8 \pm 70.5$ nM, with a Hill coefficient of $n = 1.0 \pm 0.2$. Our data reveals that the removal of the N-terminal basic domain of TRF2 resulted in a ~27-fold reduction in binding affinity, as compared to the results obtained using the same DNA hairpin sequence and full-length TRF2.

Previous research has suggested that the purified TRFH domain could aid in wrapping DNA around TRF2 to form a TRF2-DNA nucleoprotein complex, similar to a nucleosome[15]. In our study, we attempted to quantify the binding of TRFH (TRF2$_{42-245}$) using the DNA hairpin assay. However, we did not observe any delay in full unzipping of the hairpin, even at a high concentration of the protein (5 µM). This could mean that either TRFH was unable to bind to the hairpin over the concentration range tested, or its binding was too dynamic to cause a detectable delay. Regardless of the reason, the result suggests that the TRFH domain fails to stably associate with the DNA hairpin. Another explanation could be that the length of the hairpin, which is likely shorter than the ~26 nm required for effective DNA wrapping by TRFH[15], may not allow for the formation of the most stable TRFH nucleoprotein complex.

## Discussion

In conclusion, this study has shed light on crucial aspects of the interaction between full-length human TRF2 and DNA, enhancing our understanding of TRF2's role at telomeres. By

characterizing the interaction of TRF2 with DNA at the single-molecule level, we have answered several open questions regarding TRF2-DNA interactions. Most notably, our results have directly demonstrated the presence of multiple modes of TRF2's binding to DNA and shown that the binding does not exhibit a substantial preference for supercoiling chirality.

We demonstrate that TRF2 exhibits the ability to bind DNA using various modes, including a non-compact mode and multiple compact modes with varying degrees of stability. The non-compact mode is favored when subjected to increased force and/or higher salt concentrations. The existence of multiple compact binding modes of TRF2 contradicts earlier findings from AFM imaging, which reported the formation of uniform nucleoprotein units on telomeric DNA, leading to consistent length reductions of ~27 nm per unit[15]. A subsequent study by the same research group unveiled a more diverse range of nucleoprotein complexes on telomeric DNA. These complexes can be broadly categorized into two groups: a small unit where DNA traverses TRF2 with slight bending, and a larger unit where DNA appears to wrap around the edges of TRF2 proteins[14]. In our study, the notable DNA compaction step sizes observed on the 200 bp DNA fragment containing 25 repeats of the 5'-TTAGGG sequence (as shown in Fig. 1c) align more closely with the larger nucleoprotein units reported in the latter study. It's worth noting that since 25 repeats of telomeric sequences may interact with more than one TRF2 molecule, it remains unclear whether the observed multiple binding modes represent distinct conformations of a single TRF2-DNA nucleoprotein complex or complexes involving multiple TRF2 molecules.

Our findings from single-DNA supercoiling assays indicate that TRF2 binding does not exhibit a substantial preference for supercoiling chirality when DNA is subjected to tension. This result is surprising, as previous studies have suggested that TRF2 can generate positive supercoils on circular DNA in the presence of TOPO I[12]. This activity was thought to be due to TRF2's right-handed DNA wrapping. A strong right-handed wrapping activity would imply a preference for positively supercoiled DNA, but we did not observe such preference in our study. Our findings are consistent with previous crystal structures that show no change in DNA conformation upon binding to TRF2's DNA binding domain[3]. The time scale of our single-DNA supercoiling assays, which lasted from 30 min to 3 h, is comparable to typical biochemical assays and AFM imaging experiments, so the discrepancy cannot be attributed to a difference in interaction time between different assays.

This result suggests that TRF2 does not exhibit a high affinity for positively supercoiled DNA when the DNA is subjected to sub-pN tension. However, this result does not contradict previous bulk biochemical assays, which demonstrated that TRF2 can induce positive supercoiling on circular DNA in the presence of TOPO I[12]. In those experiments, DNA molecules were not subjected to the same mechanical constraints as in our study. It is possible that our assay may not be sensitive enough to detect induced chirality if it is not sufficiently strong under the applied tension. Indeed, the slight skew of the supercoiling center toward the side with higher positive supercoiling density at low forces (<0.6 pN) and with TRF2 concentrations exceeding 10 nM suggests the potential for a more pronounced chiral deformation of DNA by TRF2 as the tension approaches zero. These variations in experimental conditions may account for the previously observed generation of positive supercoils on circular DNA in the presence of TOPO I[12].

The results obtained from our single-DNA supercoiling assays are also inconsistent with the notion that TRF2 wrap DNA in both right-handed and left-handed manners, as reported for tetrasome formation (Supplementary Fig. 7b)[31,42]. If such a

scenario were true, the binding of TRF2 to DNA would not result in a shift of the supercoiling center, since the protein would not exhibit a strong preference for either right-handed or left-handed wrapping. Nevertheless, binding of such DNA wrapping proteins is expected to lead to a broadening of the parabolic cap in the linking number-extension curve[26]. Conversely, our observations show that TRF2 shifts the buckling transition points towards the supercoiling center, leading to a much narrower parabolic cap in the linking number-extension curve (Fig. 3b, c). This can be better explained by the DNA bending effect that does not cause substantial local chiral deformation. DNA bending by proteins decreases the energy barrier of DNA buckling transition and shifts the buckling transition points towards the supercoiling center, leading to a narrower parabolic cap in the linking number-extension curve, as predicted by theory[26] and demonstrated by other DNA bending proteins such as HMGA2[33].

Previous studies have reported TRF1 as a DNA bending protein[34] and capable of bringing remote DNA sites together through its long flexible linker (106 a.a.) between TRFH and Myb domains[43]. TRF2 has a similar domain organization with an additional basic N-terminal DNA binding domain and an even longer flexible linker domain (200 a.a.) between TRFH and Myb domains[43]. Therefore, it's expected that TRF2 may also have similar DNA-bending and juxtaposition activities, which may be further stabilized by the two N-terminal DNA binding domains in the TRF2 homodimer. This is supported by the $\sigma$-H data recorded from the supercoiling assay for TRF1, which shows a similarly narrowed parabolic cap (Supplementary Fig. 12). The only difference observed was a higher concentration of TRF1 required compared to TRF2, which is in line with TRF1's lower DNA binding affinity compared to TRF2.

Our DNA hairpin binding assay reveals that replacing the five telomeric repeats of 5'-TTAGGG with an equal number of non-specific DNA bases leads to a reduction in TRF2's binding affinity by ~7-fold under our solution conditions, corresponding to a difference in binding energy of ~2.1 $k_BT$. According to prior studies[44,45], the telomere specificity of TRF2's binding is mediated by its C-terminal Myb domain, and it is therefore expected that other proteins containing a Myb domain, such as TRF1, may exhibit a similar level of preference for the 5'-TTAGGG repeats. The human telomere contains 700–1800 such repeats, offering multiple specific binding sites for TRF2. Research has shown that in vivo, telomeric repeats act as sequences that are unfavored by nucleosomes[46], thus TRF2 not only has a substantially stronger binding affinity to telomeric DNA sequences, but it also faces less competition from nucleosome formation. On the other hand, on non-telomeric sequences, TRF2 experiences not only weaker binding affinity but also greater competition from nucleosome formation. These factors combine to make the telomere a highly favorable chromosomal region for TRF2 binding. The TRF2-decorated telomeres can then serve as a scaffold to interact with other shelterin proteins and shelterin-associated proteins, thereby localizing to telomeres and regulating their structure and function[47,48].

We have demonstrated that the removal of the N-terminal region (1-42) of TRF2 results in a ~27-fold reduction in its binding affinity in our experimental conditions. This indicates that the N-terminal basic domain plays a notable role in TRF2's binding affinity through non-specific electrostatic interactions between the protein and DNA. The presence of this non-specific DNA binding domain offers a cost-effective advantage, as it requires less energy and resources for protein production in cells. Importantly, this advantage does not come at the cost of specificity, as shown by a recent theoretical study[49].

It has been known that TRF2 binds DNA with multiple domains, involving the Myb domain for the sequence-specific

**Table 1 Sequences of primer oligonucleotides.**

| Handle | Primer | Sequence |
|---|---|---|
| Left handle | Forward | 5′/5Biosg/-TCA GTA CGC TAC GGC AAA TG |
| Left handle | Reverse | 5′-TTA ACC AGT TCA GTG GAG GTA TGA CAA CCA CGG AAT G |
| Right handle | Forward | 5′-TTA ACC AGG TCG GTG GTC AGT ACG CTA CGG CAA ATG |
| Right handle | Reverse | 5′-TTA ACC AGA TCC GTG GAG GTA TGA CAA CCA CGG AATG |
| Right label | Forward | 5′-TCT AGC TCT TCA AGC ATC C |
| Right label | Reverse | 5′-/5Phos/GCT TGA AGA GCT AGA/3ThioMC3-D |
| Mega left Handle | Forward | 5′-ACC TGA CGT CTA AGA AAC CAT TAT |
| Mega left Handle | Forward | 5′-ATC GCC TTG CAG CAC AT |
| Mega right Handle | Forward | 5′-CGT CGA GAT ATC GGA TGC C |
| Mega right Handle | Forward | 5′-CAT GTT CTT TCC TGC GTT ATC C |

binding, the N-terminal basic sequence to enhance the binding affinity, and a weaker TRFH domain[2], which not only further enhances the binding affinity but may also could modulate the conformation of the binding site. The TRFH domain was reported to have a DNA wrapping activity[15]. These domains work together to determine the specificity, affinity and the conformation DNA binding of TRF2[49], and could be related to the observed gradual transition from a less stable to a more stable compact conformation in Fig. 2.

A previous study reported that the N-terminal basic domain increases the DNA binding affinity of TRF2 by two-fold[18], which is notably lower than the ~27-fold change in binding affinity that we measured in our single-molecule experiments. This discrepancy could be due to differences in the solution conditions and measurement techniques used in the two studies. The previous study measured the TRF2-DNA binding using fluorescence anisotropy, where the dissociation constant was determined by fitting the measured fluorescence anisotropy $r$ with the equation $\frac{r}{r_{\max}} = \frac{c}{c + K_d}$ against the titrated protein concentration $c$. However, this approach assumes that the unbound TRF2 concentration $c$ can be approximated as the total protein concentration, which is only valid when the total protein concentration is in excess relative to the DNA concentration. In the previous study, this condition was not met as the concentration of DNA was within the range of the titrated protein concentrations.

Together, our study has addressed several key questions regarding the interactions between TRF2 and DNA, and have made substantial advancements in our understanding of TRF2-driven DNA deformations, sequence-specific binding affinities, and the role of TRF2 domains in DNA binding. Our findings suggest that a TRF2 dimer binds DNA through strong, non-specific interactions facilitated by the two N-terminal basic DNA binding domains, which are further enhanced by the sequence-specific DNA binding of the Myb domains, leading to notable DNA bending. Our results suggest the presence of multiple binding modes that may switch depending on chemical (solution) and physical (tension, supercoiling) conditions. Although our findings are limited by the experimental time scale and throughput of the single-DNA manipulation technique used, it offers high sensitivity and the ability to provide dynamic information, complementing traditional bulk biochemical assays and structural studies.

## Methods

**Building telomeric dsDNA binding sites**. The telomeric dsDNA used in the single molecule experiments was prepared by ligation of 25 repeats of the human telomeric sequence with two handles (as shown in Supplementary Fig. 1a). The telomeric dsDNA constructs were purchased from Genscript (CCATCGACATGG GAGCTC GCGTTG ATGCAT GTATGT CCCGGG TTAGGG TTAGGG TTAGGG TTAGGG TTAGGG TTAGGG TTAGGG

TTAGGG TTAGGG TTAGGG TTAGGG TTAGGG TTAGGG TTAGGG TTAGGG TTAGGG TTAGGG TTAGGG TTAGGG TTAGGG TTAGGG TTAGGG TTAGGG TTAGGG TTAGGG TTATAA GTGTTG CTGCAG CCATGTCGATGG), and later inserted into a pUC57 vector. The dsDNA contained 25 repeats of the human telomeric sequence, which were bordered by BstXI recognition sites. The plasmid was amplified in E. coli and extracted using the QIAprep Spin Miniprep kit, followed by BstXI (NEB) digestion and gel extraction to obtain the telomeric sequence with sticky ends for ligation to dsDNA handles.

To allow peeling of one strand, the left handle was labeled with 5′-biotin, while the right handle was ligated with an annealing fragment labeled with 3′-thiol. The handles were generated by PCR using bacteriophage λ-DNA and the Q5 High-Fidelity DNA Polymerase (New England Biolabs, NEB). BstXI recognition sites with non-palindromic sequences bordered the primers to generate sticky ends and prevent self-ligation. The PCR products were purified using the PureLink PCR purification kit (Invitrogen). The short fragment of the 3′-labeled thiol group on one end and the sticky end, ready for ligation with the right handle, was obtained by annealing two complementary oligos. The two sequences were mixed in equal molar amounts in annealing buffer (10 mM Tris, pH 8.0, 50 mM NaCl, 1 mM EDTA), heated to 90 °C, and cooled gradually. The four DNA pieces were then ligated by incubating them with T4 ligase overnight at 16 °C, and the final product was purified by gel extraction using the PureLink kit (Invitrogen). A similar protocol was used to build site-specific DNA hairpins. The DNA oligonucleotides were obtained from Integrated DNA Technologies (IDT) and detailed sequence can be found in Table 1.

**Building DNA hairpin**. A similar protocol is used to make a DNA construct for in situ building a hairpin as building a telomeric dsDNA binding sites, by changing the telomeric sequence to inverted DNA sequence illustrated in Supplementary Fig. 13a. The central part of DNA construct for in situ building hairpin was built by annealing two DNA oligos on each side of hairpin and making the sticky end available to ligate handles. It was followed by ligating two DNA handles the same as mentioned in the previous section to ensure labeling of biotin and thiol on the same strand. The oligos used to build a hairpin (stem 52-bp, loop 8-nt) of 5 repeats of telomeric dsDNA and the same length of hairpin with non-consensus sequence can be found in Table 2.

**Building torsional constraint DNA**. To generate torque-constrained DNA, the mega-PCR method was employed. Two DNA handles, each consisting of 510 bp, were labeled with multiple biotin and digoxigenin molecules. This was achieved by incorporating biotin-16-dUTP and digoxigenin-11-dUTP nucleotides (Roche) into the PCR reaction mix along with the dNTP solution mix. The sequences of four primers used in the

**Table 2 Sequences of hairpin oligonucleotides.**

| Hairpin | Sequence |
| --- | --- |
| 5-repeats telomeric stem | 5'-/Phos/CTTGT GCACA GACTC GTTTG GTTAT CGAAGG TTAGGG TTAGGG TTAGGG TTAGGG TTAGGG TGCTA CCAGG AAAAAAAA CCTGG TAGCA CCCTAA CCCTAA CCCTAA CCCTAA CCCTAA CCTTC GATAA CCTTT GCAGC CAGGT CAGTA GCGAC |
| Non-telomeric stem | 5'-/Phos/CTTGT GCACA GACTC GTTTG GTTAT CGAAG GTAAG GTCTG GCGAA CGGTG TATTA CCGGT TTGCT ACCAGG AAAAAAAA CCTGG TAGCA AACCG GTAAT ACACC GTTCG CCAGA CCTTA CCTTC GATAA CCTTT GCAGC CAGGT CAGTA GCGAC |
| Linker Primer Forward | 5'-/Phos/CTA CTG ACC TGG CTG C |
| Linker Primer Reverse | 5'-CGA GTC TGT GCA CAA GGT GC |

**Table 3 PNA oligos.**

| PNA # | Sequence |
| --- | --- |
| 1 | 5'-AGG TAT GAC AAC CAC GGA ATG CAT TTT TCT |
| 2 | 5'-GGC AGC GGG CTT CAT ATT CTG TGT GCT TAT |
| 3 | 5'-GCT TGC CGA CAT GGG ACT TGT TCA ATG ACA |
| 4 | 5'-CCT CAG CAG GAA AAC GCC CTT CGC AGC ATT |
| 5 | 5'-GCC CGT CAG GCT AAT TCT GAA ATC AAA AAA |
| 6 | 5'-AGC CAG ACA GCA GTT TCC GGA TAA AAA CGT |
| 7 | 5'-CGA TGA CAT TTG CCG TAG CGT ACT GA |
| 8 | 5'-CAA CGC GAG CTC CCA ATT CAG TGG |
| 9 | 5'-CCA CCG ACA TGG |
| 10 | 5'-TCTAGCTCTTCAAGCATCC GTGG |

reaction can be found in Table 1. Subsequently, the biotin- and digoxigenin-labeled handles were mixed with the bacteriophage-$\lambda$ DNA template and subjected to PCR using Q5 Hot Start polymerase (NEB). All PCR products, including the ones mentioned above, were purified using the PCR Purification Kit (ThermoFisher).

**Form DNA tethers**

*DNA hairpin tethers.* The DNA sequence is designed to fold into a hairpin structure after one complementary strand is removed (Supplementary Fig. 13a). Initially, the tethered DNA is double-stranded (dsDNA), with one strand tethered between the super-paramagnetic bead and the coverslip surface. Subsequently, we apply a force exceeding the overstretching threshold under low-salt conditions to induce dissociation between the strands, resulting in a single-stranded DNA (ssDNA) tethered between the bead and the surface. This tethered ssDNA contains an inverted sequence that can form a hairpin structure when the force decreases. The remaining single-stranded DNA regions on either side of the hairpin are blocked using PNA. The detailed sequence of the PNA can be found in Table 3.

*DNA containing 25 telomere repeats under tension.* A similar approach based on overstretching was employed to create a specific site under tensile force (Supplementary Fig. 1a). In this scenario, the tethered ssDNA, following the overstretching transition, lacks an inverted sequence, preventing the formation of a hairpin structure at low forces. Instead, it consists of 25 repeats of the sequence 5'-TTAGGG. To construct the specific telomere dsDNA site, a complementary strand containing 25 repeats of 5' CCCTAA was introduced into the chamber. Following annealing, a dsDNA region with 25 repeats of 5'-TTAGGG was generated. In a similar manner, PNA was introduced to block the spanning ssDNA regions. For a more detailed methodology, please refer to our previous publication[39], where we have provided a comprehensive description of how the DNA is prepared and manipulated.

*Torsional constrained DNA for supercoiling assay.* Purified 6.5 kbp DNA molecules were introduced into the chamber, with multiple biotin labels at one end and multiple digoxigenin labels at the other end to create tethers. DNA molecules that were torsionally unconstrained, for instance those with nicks, were precluded from the study based on their lack of responsiveness to changes in linking number at forces below 0.7 pN.

**Protein purification**. DNA oligonucleotides that were used to generate the different constructs of human TRF2 were ordered from IDT. Full-length human TRF2 construct (1-500 aa) in pET30a vector was kindly provided by Dr. Daniela Rhodes. Four version of TRF2 was expressed and purified. This included the full-length human TRF2 (hTRF2, 1-500 aa); N-terminally truncated TRF2 ($\Delta$BTRF2, 45-500 aa) lacking the N-terminal basic domain; and TRF2 homodimerization domain alone (TRFH, 42-245 aa) lacking the N-terminal basic domain as well as the myb DBD domain (Supplementary Fig. 15a). The two truncated TRF2 constructs ($\Delta$BTRF2 and TRFH) were generated by site-directed deletion of the pET30a full-length TRF2 using PCR. Full-length human TRF2 and truncated TRF2 mutants were expressed in BL21 Rosetta competent cells (Novagen) as N-terminal hexahistidine fusion constructs at 15 °C. The fusion proteins were purified by metal-affinity chromatography with Ni-NTA agarose resin (Qiagen) and Superdex 200 size exclusion chromatography (GE Healthcare) followed by overnight cleavage of the N-terminal hexahistidine tag with tobacco etch virus (TEV) protease at 22 °C. After TEV digestion another round of metal-affinity chromatography with Ni-NTA agarose resin was performed to remove the his-tagged TEV protease and any un-cleaved hexahistidine TRF2 protein. Finally, the proteins were loaded on a Superose 6 size-exclusion column pre-equilibrated in a buffer containing 25mM HEPES-Na pH 7.5, 150mM NaCl, 1mM DTT to obtain pure protein. Supplementary Fig. 15d shows the purity of the different TRF2 proteins used in the experiments. The elution profiles of the various TRF2 constructs used in this study are shown in Supplementary Fig. 15c. The experimental molecular weights obtained from gel filtration chromatography are similar to the expected molecular weights of the proteins.

**Single DNA stretching experiments**. A flow chamber with a volume of 10-20 $\mu$L was built on a (3-Aminopropyl)triethoxy silane (APTES) functionalized coverslip (32 × 24 mm) (Sigma-Aldrich). The thiol end of the DNA was covalently attached to the amine group of the APTES using a sulfo-SMCC crosslinker (Thermo Scientific). The APTES coverslip was first treated with 1 mg/mL of sulfo-SMCC (dissolved in 1 × PBS buffer with pH 7.4) for 30 min. After washing out the unbound sulfo-SMCC, the thiol-labeled DNA construct was added to the chamber and incubated for another 30 min. The chamber was then blocked with a BSA solution (10 mg/mL BSA, 1mM 2-mercaptoethanol, 1 × PBS buffer with pH 7.4) for over 2 h prior to experiments.

Once the DNA constructs were attached to the surface, 2.8 μm-diameter streptavidin-coated superparamagnetic beads (Dynal M-280, Life Technologies) were introduced to bind to the biotin end of the DNA. Finally, the buffer was changed to the assay buffers for single-molecule stretching experiments.

A vertical magnetic tweezers setup built in the lab was used to stretch the DNA constructs using a pair of magnets on the top of the chamber. The setup was controlled using a LabVIEW program (National Instruments) written in-house. A rotation stage (DT-50, Physik Instruments) was used to wind/unwind torsion-constrained DNA tethers by rotating the magnet pair in clockwise or counter-clockwise direction. The extension change of the construct was recorded with a sampling rate of ~200 frames per second. Force was controlled by changing the distance between the permanent magnets and bead attached to the DNA. The magnetic tweezers have a spatial resolution for bead stuck on surface of ~2 nm, and the force calibration has a relative error of <10%[16]. All experiments were done in 20 mM HCl-Tris (pH 7.4), 150 mM KCl, 1 mM MgCl$_2$ at $24 \pm 1$ °C in celsius unless otherwise stated.

The magnetic tweezers record the height change of the bead at various applied forces, which is referred to as the Force-Height (F-H) curve in this study. The height change of the bead can be a result of both molecular extension change and bead rotation due to torque rebalance. However, at a constant force or during a stepwise extension change with a specific rate, the height change of the bead reflects the extension change of the molecule[17]. The bead height is relative to a reference, which is usually taken as the extension of the original double-stranded DNA tether at 2 pN as predicted by the Worm-Like Chain (WLC) polymer model, using the Marko-Siggia formula, with a bending persistence length of 50 nm[50].

**Single DNA supercoiling assay**. In the single-DNA supercoiling assays, both strands of DNA were tethered at one end to the coverslip surface and at the other end to a superparamagnetic bead, allowing the linking number (Lk) of the DNA to be controlled by rotating the bead[23]. The superhelical density of the DNA, defined as $\sigma = \frac{\Delta Lk}{Lk_0}$, where $\Delta Lk = Lk - Lk_0$ and $Lk_0 = 6500/10.4 = 625$ is the linking number for a relaxed B-form DNA[25,51], is a length-independent description of the deviation of the DNA's linking number from the relaxed B-form. The linking number can be altered by rotating the bead and $\Delta Lk$ is equal to the number of bead rotations divided by $2\pi$.

**Reporting summary**. Further information on research design is available in the Nature Portfolio Reporting Summary linked to this article.

## Data availability

The source data for the graphs in the main figures are available as Supplementary Data in an Excel file. Data in the Supplementary Notes are available from the corresponding authors upon request.

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

## Acknowledgements

The research was funded by the Singapore Ministry of Education Academic Research Funds Tier 2 (MOE-T2EP50220-0015) and the Ministry of Education under the Research Centres of Excellence programme (J.Y.), Nanyang Institute of Structural Biology (D.R. and S.S.), and Singapore Ministry of Education Academic Research Fund Tier 3 [MOE2012-T3-1-001] (D.R. J.Y., S.S.).

## Author contributions

X.Z., M.L. and Y.Z. performed the single-molecule experiments and analyzed the data. V.K.V. purified the different TRF2 mutants, provided plasmid containing telomeric DNA repeats, performed the gel electrophoretic mobility assay and the electron microscopy imaging experiments; J.Y., S.S. and D.R. conceived and supervised the research; X.Z. and J.Y. wrote the manuscript. X.Z. and V.K.V. are co-first authors to this paper.

## Competing interests

The authors declare no competing interests.
