## [Peer Review File · Communications Biology]

Reviewers' comments:

Reviewer #1 (Remarks to the Author):

As far as I can tell, the authors do not provide any new insights into how TRF2 interacts with telomeric DNA. Although some of the approaches are new, the results are not. The only new result may be that the presence of the Basic domain enhances DNA binding by a fair amount. Such enhancement was observed by others but the magnitude is different. It is very well possible that this difference is due to the substrate used (hairpin flanked by ssDNA) since the Basic domain has a preference for branched DNAs regardless of sequence.

Reviewer #2 (Remarks to the Author):

In this manuscript Zhao et al. have investigated the binding properties of the telomeric protein TRF2 and some truncations of this protein on different forms of DNA by magnetic tweezers. Although the results obtained are novel and experiments are correctly executed, the experimental settings and the presented data raise several questions of interpretation that require an in-depth reevaluation of the conclusions. In particular, as detailed below, by no means the presented data allow the authors to negate the existence of DNA wrapping and chirality properties of TRF2 as it is stated in the title.

1) A key question regarding the interpretation of the presented results is whether the authors analyzed a TRF2-DNA complex that has reach its final form and, above all, a form corresponding to the one adopted in solution where no force is applied.

Indeed, the authors state that the fact that TRF2 has the same behavior in experiment performed at 0.35 pN with 60 sec steps or 10 seconds steps suggest near equilibrium conditions (page 4 line 380 to 386). While in page 3 they state that at force as low as 0.3 pN it takes more than 500 s to reach a more stable complex (lane 256 to 263).

Hence, at low force the system fluctuates and takes more than 500 sec to adopt its final compaction. Therefore, the 10 (or 20 seconds) seconds time points and the force applied (from 0.35 pN to 1.3 pN) of the following experiments do not allow to conclude about a lack of chirality preference. It is highly likely that the authors are observing some intermediate states formed at high force. Thus, the topology of the complex when applying low force or without force (in solution) and when the equilibrium is reached is not investigated in this study.

This explains why TRF1 has similar trace. TRF1 cannot wrap DNA because of an acidic N-terminus and thus behaves like TRF2 when one applies a force.

Thus, it is a clear misinterpretation for the authors to negate the event of DNA wrapping while analyzing the complexes in conditions unsuitable for detecting it and furthermore, ignoring a previous publication showing that TRF2 does prefer positively supercoiled DNA (reference 1) and has therefore a chirality preference.

2) However, an interesting observation can be made from the presented data. On Figure 3, at low force the hysteresis is only present for relaxed and positively supercoiled (sc) DNA suggesting that the complexes take more time to form in these conditions and are thus of a different nature than the ones formed with negative sc DNA. One possibility is that, in this case, a second equilibrium takes places with lower kinetics. This stresses again the necessity to use longer time points and low force in all experiments or failing that to point out that the complexes they investigate are early forms of complexes formed when force is applied.

3) The authors suggest (page 6 from line 532) that the TRFH domain does not bind DNA or in a very dynamic manner just after admitting that their hairpin substrate could be totally unsuitable to observe binding. Indeed, the TRFH domain needs ~90 bp of linear DNA to bind efficiently (reference 2). Thus, it is not surprising that the authors do not see a binding on a 52 bp hairpin. This part should be removed since the authors do not show the data and, above all, contradict themselves.

4) The authors should consider published results showing that TRFH is a weak DNA binder (reference 2). In the event of TRF2 binding on DNA thanks to the Myb/SANT domain, the TRFH is

thus in close proximity to be able to bind, likely through a second equilibrium slower than the initial binding. This explains why applying force and looking at short time points preclude observing chirality preference. This also explains the difference with nucleosomes which have a very high affinity and can sustain high forces.

5) The authors show a gel with loss of binding when glutaraldehyde is added. Glutaraldehyde has been used for many years in electron microscopy and AFM to study many different DNA-protein complexes including nucleosomes (examples in references 3 and 4) and TRF2 (reference 5). Originally, the use of Glutaraldehyde was introduced and validated for electron microscopy. Indeed, it was shown that the structural and physical properties of nucleosomes were not perturbed by this crosslinking (references 6, 7 and references therein). That led to the use of this compound for studying different looped structures (references 8 to 10) and in AFM in the studies of chromatin (references 11 to 13 for examples) and later on for TRF2 itself in both techniques (references 5 and 14). In all these experiments, TRF2 experiments included, no report of any modified binding has been made. Moreover, some key AFM experiments showing the wrapping (DREEM) were performed without glutaraldehyde and after 20 minutes of incubation. These latter experiments show without doubt the wrapping since one can see the DNA around the protein (reference 2), again underlining the fact that experimental conditions in the present manuscript are unsuitable for studying TRF2 wrapping/chirality preference.

References

- 1- TRF2 and apoII cooperate with topoisomerase 2alpha to protect human telomeres from replicative damage.
Ye J, Lenain C, Bauwens S, Rizzo A, Saint-Léger A, Poulet A, Benarroch D, Magdinier F, Morere J, Amiard S, Verhoeyen E, Britton S, Calsou P, Salles B, Bizard A, Nadal M, Salvati E, Sabatier L, Wu Y, Biroccio A, Londoño-Vallejo A, Giraud-Panis MJ, Gilson E. *Cell*. 2010 Jul 23;142(2):230-42. doi: 10.1016/j.cell.2010.05.032. PMID: 20655466
- 2- Benarroch-Popivker, D., Pisano, S., Mendez-Bermudez, A., Lototska, L., Kaur, P., Bauwens, S., Djerbi, N., Latrick, C.M., Fraissier, V., Pei, B. et al. (2016) TRF2-Mediated Control of Telomere DNA Topology as a Mechanism for Chromosome-End Protection. *Mol Cell*, 61, 274-286.
- 3-Wang H, Bash R, Yodh J, Hager G, Lindsay S, Lohr D. (2004) Using atomic force microscopy to study nucleosome remodeling on individual nucleosomal arrays in situ. *Biophysical journal*.;87(3):1964–1971.
- 4-Caño S, Caravaca JM, Martín M, Daban JR. (2006) Highly compact folding of chromatin induced by cellular cation concentrations. Evidence from atomic force microscopy studies in aqueous solution. *European Biophysics Journal*.;35(6):495–501.
- 5-Shige H. Yoshimura, Hugo Maruyama, Fuyuki Ishikawa, Rieko Ohki and Kunio Takeyasu (2004) Molecular mechanisms of DNA end-loop formation by TRF2 *Genes to Cells* 9,205 – 218.
- 6-Involvement of histone H1 in the organization of the nucleosome and of the salt-dependent superstructures of chromatin
J Cell Biol. 1979 Nov 1; 83(2): 403–427.
THOMA,TH.KOLLER,and A.KLUG
- 7-Conformational characterization of nucleosomes by principal component analysis of their electron micrograph
M. M. Z. ZABAL, GJ CZARNOTA, D. P. BAZETT-JONES, F. P. OTTENSMEYER
Journal of Microscopy, Vol. 172, Pt 3. December 1993, pp. 205-214.
- 8-Tatiana Nikitina, Christopher L. Woodcock
jcb Home » 2004 Archive » 19 July » 166 (2): 161

9-Single-molecule compaction of megabase-long chromatin molecules by multivalent cations
Anatoly Zinchenko, Nikolay V Berezchnoy, Sai Wang, William M Rosencrans, Nikolay Korolev, Johan R C van der Maarel, Lars Nordenskiöld
Nucleic Acids Res. 2018 Jan 25; 46(2): 635–649.

10-Nucleosome-like, Single-stranded DNA (ssDNA)-Histone Octamer Complexes and the Implication for DNA Double Strand Break Repair
Nicholas L. Adkins, Sarah G. Swygert, Parminder Kaur, Hengyao Niu, Sergei A. Grigoryev, Patrick Sung, Hong Wang, Craig L. Peterson
J Biol Chem. 2017 Mar 31; 292(13): 5271–5281. Published online 2017 Feb 15. doi: 10.1074/jbc.M117.776369

11- Chromatin fiber structure: morphology, molecular determinants, structural transitions.
Biophys J. 1998 May;74(5):2554-66.
Zlatanova J1, Leuba SH, van Holde K.

12- Contributions of linker histones and histone H3 to chromatin structure: scanning force microscopy studies on trypsinized fibers.
Biophys J. 1998 Jun;74(6):2823-9.
Leuba SH1, Bustamante C, Zlatanova J, van Holde K.

13-Human centromere protein A (CENP-A) can replace histone H3 in nucleosome reconstitution in vitro
Kinya Yoda, Satoshi Ando, Setsuo Morishita, Kenichi Houmura, Keiji Hashimoto, Kunio Takeyasu, and Tuneko Okazaki
PNAS June 20, 2000 97 (13) 7266-7271; <https://doi.org/10.1073/pnas.130189697>

14-The basic domain of TRF2 directs binding to DNA junctions irrespective of the presence of TTAGGG repeats.
Fouché N, Cesare AJ, Willcox S, Ozgür S, Compton SA, Griffith JD.
J Biol Chem. 2006 Dec 8;281(49):37486-95. Epub 2006 Oct 18.

Reviewer #3 (Remarks to the Author):

I read with interest the work Zhao et al entitled "TRF2 shows no detectable preference for chirality of supercoiled DNA and no 2 wrapping of telomeric DNA in single-DNA manipulation studies".

This work focusses on DNA TRF2 interaction at the single molecule level with a mechanical approach using on magnetic tweezers. The authors measure interaction on dsDNA in a linear relaxed form to detect its effect on such a DNA, in a hairpin form to measure the affinity of the wild type and of variants on its target sequence or on a different DNA sequence, and more importantly on a supercoiled DNA as an asymmetric binding of TRF2 was published in a previous study.

The paper tests 3 points on 3 types of DNA substrates.

Starting from the linear relaxed DNA to detect DNA extension modification by TRF2:

-There is no comment on the size of the shortening. 50nm is quite large. It could be compared to the size of the binding region ($25 \times 6 \times 0.34 = 51 \text{nm}$! that could eventually be corrected by an extension factor related to a WLC to consider the force) or to a footprint size of the enzyme. The proximity of the 2 distances (step size and size of the target sequence) brings confusion. A longer/shorter size of repeats would give a different value? This is to understand if the shortening measured is structural or related to the size of the target DNA that would in a way "collapse". As

mentioned in the text there could be interaction between different bound TRF2. Why not using a DNA construct with single or few spaced TTAGGG as, demonstrated in the "hairpin" section the affinity for this telomeric sequence is much higher than on random DNA? It would be important in view of the arguments developed about the different step sizes in the histogram (lines about 211). In addition with twice the minimum step size the PNA/DNA handles should be involved in the deformation. It is not a heavy work and it should clarify points on the detected states.

-The time scale for the fluctuations between the different states do not depend on the force? Usually such fluctuations and their variations with force can bring interesting information on the energy landscape. This was often used in this field. Nothing is used here. Is there any reason? It is the non equilibrium aspect appearing in the F-H curves that prevented to do that? The fig2 pushes to deepen this aspect.

-A new element brought by this study is the existence of a more stable compact complex after some time as demonstrated in fig2. How long the time mentioned ($> 500s$) compares with the time between the injection and the start of the F-H curves? Those F_H curves present an hysteresis but are the curves almost the same after and before the appearance of this "stable" and compact state? When it appears, this stable state last forever at low force? How long did the authors wait ($500s$ being already quite long) ? This state brings a question on all the timescales of the experiments as underlined at different point in the text. Also in hairpin part, a more stable conformation would take longer to be removed, is this observed? Nevertheless this more stable state is interesting and should be investigated in future studies.

-Line 269 the three states should be more clearly defined. If I am correct they correspond to the extended DNA form, then 2 short extensions forms but one being more stable than the other? This paragraph lacks some clarity for me.

-At the end of this part I am confused by the remark of the salt increase that shuts down the detection of the interaction while TRF2 remains on the DNA. It should clearly be more commented (it seems to imply at least 2 contact points one suppressed in the presence of high salt concentration). It raises the question TRF2 binding on ssDNA as this could affect the detection of the hairpin part if TRF2 could stay on the ssDNA. Do the authors have some elements on this point?

Supercoiled DNA tod detect a bing dependant on the sign of the supercoiling :

-I start with a minor point in figure 3 symbols should be clarified as they are in fig 1. It is currently in the text but it would be better in the legend of the figure. Adding the force as insets in the figure would be nice too for non experts of magnetic tweezers.

-It is not clear whether the DNA template has the telomeric sequence. It is mentioned non-specific DNA so I would guess it is not the case but it should be the case for me, the specific binding to the telomeric sequence, and its chirality may depend on the sequence. The same experiment could be performed on a part of the cloned PUC57 used to prepare the linear construct.

-Indeed the proteins do not modify the curves presented in an asymmetric way. Nevertheless, at high force for $\sigma < 0$ the denatured or L-DNA disappears, this implies that the DNA buckles before denaturation at the same force. At what force it reappears? A force that prevents binding of TRF2? Could the authors comment?

-Why not using the same TRF2 concentrations as in fig 1? In particular high concentrations to cover the DNA as much as possible?

-Finally on this part , I think an important measurement is missing: an extension vs time in the presence of supercoiling to detect the binding of individual TRF2 in a way similar to the work detailed in "Promoter unwinding and promoter clearance by RNA polymerase: Detection by single-molecule DNA nanomanipulation" by the group of T. Strick in PNAS. I think it was the first paper that used this approach but others have done since then. If any, differences in the binding/unbinding kinetics could bring differences in the affinity but also potential differences in the change of height induced by binding on the two sides could help deconvolute the bending and

torsional components. These measurement would be more sensitive than the curves presented here I think. These would bring more complete proofs to a major claim of the paper.

The present conclusion on this part, in contradiction to previous results as noted in the manuscript but with a discrepancy not justified, should be more robustly argued. Could the authors reproduce the experiment with TOP1 in bulk with their protein?

Hairpin part to measure the affinity of TRF2 to its specific site and the parts of the enzyme involved in binding:

- I think it would be worthwhile for non-specialists to explicitly indicate that there no force in the hairpin.

-The addition of non specific (10 and 12 bases) DNA outside of the target site have no clear reason, for me, to be here in relation to the refolding of the hairpin. Can the authors give more elements? The authors use more than one telomeric sequence why not using the same 25 repeats as in the linear part? In any case a longer DNA hairpin would avoid the remark made line 544. And the construct is not much more complex.

-I have a technical question : why having a minimum of only 50 cycles, which is limited, and run bootstrap rather than using more statistics with longer acquisition times as the experiment seems to be self running and one cycle is not so long (with again the remark of an experiment where the protein could bind for more than 500s as mentioned in the linear DNA part)?

-Here again there are published approaches with a similar setup and configuration that could bring more quantitative information on a similar experiment. In "Single-molecule kinetic locking allows fluorescence-free quantification of protein/nucleic acid binding" Rieu et al. used a similar hairpin opening approach but that could bring kinetics elements in addition to the thermodynamic one as in the present paper. Why not using this approach? Does TRF2 precludes this? Timescales are too long? Nevertheless, to be more positive , the test on the effect of the DNA sequence and TRF2 domains binding properties is rather convincing if one excludes the possibility of a too short hairpin affecting the measure (see my previous remark).

Minor general remarks :

-Bibliography should be checked as it seems to be victim of a latex bad configuration (DNA in lowercase being the most typical). And there are few typos (in the supp mat also).

-Construct part in the protocol main text is surprisingly not so clear. A figure in the supp mat would help.

My general conclusion is that the work is reasonable, part of the conclusions (binding thermodynamics and partially the detection on linear DNA) are correctly justified but the supercoiling part, emphasized in the title, should be more supported and other parts require also complementary experiments. So the work should be improved to reach the point where a publication could be envisioned.

We thank the reviewers for the comments that have helped us to revise and improve the manuscript. All the comments from the reviewers have been addressed. Please find below the point-to-point replies to the comments. The list of main changes is added after the point-to-point replies. The changes in the main text are marked in red.

Responses to reviewers

Reviewer #1 (Remarks to the Author):

Comment #1:

As far as I can tell, the authors do not provide any new insights into how TRF2 interacts with telomeric DNA. Although some of the approaches are new, the results are not. The only new result may be that the presence of the Basic domain enhances DNA binding by a fair amount. Such enhancement was observed by others but the magnitude is different. It is very well possible that this difference is due to the substrate used (hairpin flanked by ssDNA) since the Basic domain has a preference for branched DNAs regardless of sequence.

Response #1:

{

We disagree with the comment stating, "The only new result may be that the presence of the Basic domain enhances DNA binding by a fair amount." One of our main findings, the lack of strong chiral preference in TRF2 binding and its weak dependence on the effect, is a novel and unexpected discovery. Additionally, the multi-stage binding modes, including the initial highly dynamic compaction of DNA and the subsequent stable DNA compaction, were not reported in previous studies. Furthermore, our approach to the supercoiling assay of TRF2 is also innovative and novel.

Addressing the last comment, which suggests that the difference in results may be attributed to the substrate used (hairpin flanked by ssDNA) and the Basic domain's preference for branched DNAs regardless of sequence, we want to emphasize that our hairpin was not flanked with ssDNA handles. Instead, it was spanned between a PNA/ssDNA duplex to prevent binding to branched DNAs. The PNA/ssDNA duplex proved to be highly effective, allowing us to quantify the impacts of the basic N-terminal domain and the sequence-dependent binding of TRF2. If the PNA/ssDNA handles did not adequately suppress TRF2's branch-binding, we would not have observed these effects.

To further support this, we demonstrated that using ssDNA handles to span the same hairpin resulted in strong TRF2 binding that completely prevented the unzipping of the DNA hairpin at the probing force (Figure R1 B). This is in sharp contrast to the transient pauses observed during the unzipping of the DNA hairpin when using PNA/ssDNA handles (Figure R1 A). This result provides additional evidence that the PNA/ssDNA effectively suppresses TRF2's branch-binding. This suppression can be attributed to the non-native conformation of the PNA/ssDNA handles, which is unfavorable for TRF2 binding.

Figure R 1 Representative time trace for TRF2 binding on DNA hairpin flanked by PNA/ssDNA duplex handles (A) and ssDNA handles (B), respectively. (C) No apparent deformation for TRF2 binding to 600 bp PNA/ssDNA up to 200 nM. (D) TRF2 compact 1000 nt ssDNA as concentration increasing to 50 nM.

We have presented additional experimental evidence demonstrating the absence of detectable binding between TRF2 and the PNA/ssDNA duplex (Figure R1 C). Specifically, on a 600 bp PNA/ssDNA duplex, we did not observe any extension changes induced by TRF2 binding, in contrast to the TRF2-mediated compaction of dsDNA. Furthermore, we conducted experiments using 1000 nt ssDNA and observed a distinct TRF2-dependent shift in the force-extension curve (Figure R1 D). This finding indicates the presence of an ssDNA-binding activity of TRF2, which can account for the branch-binding activity of TRF2 when the hairpin is bridged between ssDNA handles, as depicted in Fig. R1B.

We have provided the following sentence in the second paragraph in Results, “The PNA/DNA hybrid handles were introduced used to suppress non-specific binding of TRF2 to the handles or to the fork of the hairpin, which were proven to be highly effective” and linked it to SI Fig. S1 and S13 for experimental evidence.

}

Reviewer #2 (Remarks to the Author):

In this manuscript Zhao et al. have investigated the binding properties of the telomeric protein TRF2 and some truncations of this protein on different forms of DNA by magnetic tweezers. Although the results obtained are novel and experiments are correctly executed, the experimental settings and the presented data raise several questions of interpretation that require an in-depth reevaluation of the conclusions. In particular, as detailed below, by no means the presented data allow the authors to negate the existence of DNA wrapping and chirality properties of TRF2 as it is stated in the title.

Comment #2:

1) A key question regarding the interpretation of the presented results is whether the authors analyzed a TRF2-DNA complex that has reached its final form and, above all, a form corresponding to the one adopted in solution where no force is applied. Indeed, the authors state that the fact that TRF2 has the same behavior in experiment performed at 0.35 pN with 60 sec steps or 10 seconds steps suggest near equilibrium conditions (page 4, line 380 to 386). While in page 3 they state that at force as low as 0.3 pN it takes more than 500 s to reach a more stable complex (line 256 to 263).

Hence, at low force the system fluctuates and takes more than 500 sec to adopt its final compaction. Therefore, the 10 (or 20 seconds) seconds time points and the force applied (from 0.35 pN to 1.3 pN) of the following experiments do not allow to conclude about a lack of chirality preference. It is highly likely that the authors are observing some intermediate states formed at high force. Thus, the topology of the complex when applying low force or without force (in solution) and when the equilibrium is reached is not investigated in this study.

This explains why TRF1 has similar trace. TRF1 cannot wrap DNA because of an acidic N-terminus and thus behaves like TRF2 when one applies a force.

Thus, it is a clear misinterpretation for the authors to negate the event of DNA wrapping while analyzing the complexes in conditions unsuitable for detecting it and furthermore, ignoring a previous publication showing that TRF2 does prefer positively supercoiled DNA (reference 1) and has therefore a chirality preference.

Response #2:

{

The reviewer referred to the experiments presented in Figure 2, where a DNA molecule was held at a sub-piconewton force that caused it to rapidly switch between a compact and an extended state for hundreds of seconds before reaching a more stable compact state. This was indicated by the absence of the "unfolding" transition. The time fractions of the compact state in the representative time traces of Figure 2 were only around 10% of the total time before reaching the stable state. Therefore, the actual time allowed for the complex to search for the stable state could be much shorter, within minutes.

Although the nature and physiological relevance of this highly dynamic state of the TRF2-DNA complex are unclear, its existence over a time scale of minutes warrants further investigation. TRF2 not only binds DNA but also interacts with other telomere proteins. A dynamic initial state of the TRF2-DNA complex may be beneficial for the bound TRF2 to recruit other telomere proteins to form the shelterin complex.

Regarding whether the more stable TRF2-DNA complex can wrap DNA with chiral preference, we have conducted new supercoiling experiments over different time scales. However, our new data presented in Fig. R2 below, which is also included as a new Supplementary Figure (Fig. S8), still do not reveal a detectable chiral preference of DNA binding by TRF2. Although these data do not completely exclude the possibility of right-handed DNA wrapping activity of TRF2, they do not support strong chiral preference of DNA wrapping by TRF2 (page 7, lines 665-672).

Briefly, in the new supercoiling assay experiment, we changed the DNA linking number ΔLk from the relaxed DNA by (+) or (-) 20 at a fixed force of 0.3 pN and held the DNA for different durations in the presence of 20 nM TRF2. The binding of TRF2 to the supercoiled DNA would result in reduced extension after rapid relaxing the supercoiling constraint of the DNA back to $\Delta Lk = 0$. Should TRF2 binding have a chiral preference, we would anticipate different extensions of the DNA between (+) and (-) supercoiling constraints.

Fig. R2 (Fig. S8 in Supplementary) shows the bead height changes during (+) and (-) ΔLk changes in the absence (Fig. R2A) or presence (Fig. R2B) of TRF2 during cycles of force change and linking number change described below. Each cycle started from the supercoiling-relaxed state ($\Delta Lk = 0$) at a high force of 8 pN for 60 s, then a state of $\Delta Lk = 0$ and 0.3 pN for 60 s, and then a state of $\Delta Lk = 20$ or $\Delta Lk = -20$ at 0.3 pN for various durations to allow TRF2 binding to supercoiled DNA. The higher force of 8 pN was introduced to displace TRF2 proteins bound on the plectonemes in the previous cycle. The various durations at the state of $\Delta Lk = 20$ or $\Delta Lk = -20$ at 0.3 pN were introduced to probe potential time-dependent chiral binding of TRF2.

In 20 nM TRF2, in each cycle, the bead height after supercoiling-relaxation ($\Delta Lk = -20$, 0.3 pN) was recorded. Its difference from the bead height of the same DNA at the same supercoiling-relaxed state ($\Delta Lk = -20$, 0.3 pN) in the absence of TRF2 is plotted as a data point Fig. R2C. As demonstrated by the data in Fig. R2C, there is no statistically significant difference in TRF2 binding to (+) or (-) supercoiled DNA regardless of the duration of the supercoiled state up to 600 s.

Figure R2 (Fig. S5). Time dependent of TRF2 binding on supercoiled DNA. (A) DNA tethers were first held at 0.3 pN and twisted to $\Delta Lk = +20$ to form positive supercoiled DNA and to $\Delta Lk = -20$ to form negative supercoiled DNA, respectively. The formed supercoiled DNA remained for designed time windows (60, 300, 600 seconds). After that, the tethered DNA was twisted back to $\Delta Lk = 0$. (B) In the presence of TRF2 20 nM, slower relaxation and shorter extension of bead height was observed compared to sharp bead extension jump in (A) when the supercoiled DNA twisting back to $\Delta Lk = +20$. (C) Summary of the bead height deduction caused by TRF2 binding, revealing no detectable chiral preference of DNA binding by TRF2.

While we did not observe DNA supercoiling by TRF2 in these assays, these data do not rule out the possibility that TRF2 might have a weak preference for torsion-relaxed positively supercoiled DNA. There is a possibility that our assay might not be sensitive enough to detect if the induced chirality is not strong enough. We also note that the DNA in this single-molecule assay is under sub-pN forces, which is different from previous bulk assay utilizing circular DNAs that are not under similar mechanical constraint. This variation in experimental conditions could account for the previously observed generation of positive supercoils on circular DNA when TOPO I is present. These sentences have been included in Discussion section (page 7, lines 664-675).

}

Comment #3:

2) However, an interesting observation can be made from the presented data. On Figure 3, at low force the hysteresis is only present for relaxed and positively supercoiled (sc) DNA suggesting that the complexes take more time to form in these conditions and are thus of a different nature than the ones formed with negative sc DNA. One possibility is that, in this case, a second equilibrium takes place with lower kinetics. This stresses again the necessity to use longer time points and low force in

all experiments or failing that to point out that the complexes they investigate are early forms of complexes formed when force is applied.

Response #3:

{

We believe that the data presented in Figure 3 do not demonstrate that TRF2 binding causes hysteresis for relaxed DNA. The term "relaxed" in panel 3A refers to DNA with zero linking density ($\Delta Lk/Lk_0$) in the absence of TRF2.

Figure 3B, C, and D display supercoiling data at three distinct forces: 0.3 pN, 0.6 pN, and 1.3 pN, respectively, and at three TRF2 concentrations. Hysteresis was observed in both negatively and positively supercoiled DNA in Figure 1B at 10 nM and 15 nM TRF2 concentrations. Our findings do not indicate a significant difference between the effects of negative and positive supercoiling on TRF2 binding at low forces.

This observation is consistent with the new data acquired during the revision process (aforementioned Figure R 2C), where no significant difference between positive and negative linking number changes is observed.

}

Comment #4:

3) The authors suggest (page 6 from line 532) that the TRFH domain does not bind DNA or in a very dynamic manner just after admitting that their hairpin substrate could be totally unsuitable to observe binding. Indeed, the TRFH domain needs ~90 bp of linear DNA to bind efficiently (reference 2). Thus, it is not surprising that the authors do not see a binding on a 52 bp hairpin. This part should be removed since the authors do not show the data and, above all, contradict themselves.

Response #4:

{

We agree with the suggestion. Accordingly, we have removed the sentences pertaining to DNA binding by the TRFH domain. We have retained the other data, since the length of five repeats of TTAGGG sequence is sufficient for Myb-like domain to bind (Court et al. 2005).

}

Comment #5:

4) The authors should consider published results showing that TRFH is a weak DNA binder (reference 2). In the event of TRF2 binding on DNA thanks to the Myb/SANT domain, the TRFH is thus in close proximity to be able to bind, likely through a second equilibrium slower than the initial binding. This explains why applying force and looking at short time points preclude observing chirality preference. This also explains the difference with nucleosomes which have a very high affinity and can sustain high forces.

Response #5:

{

We concur with the comment. The reviewer alludes to a multivalent interaction between TRF2 and DNA that involves Myb for the initial sequence-specific binding, as well as the weaker TRFH domain, which not only enhances the binding affinity but may also modulate the conformation of the binding site. Furthermore, the N-terminal basic sequence plays a significant role in these multivalent interactions. Such multivalent interactions could drastically boost binding affinity without sacrificing the binding specificity, as discussed in our recent publication (Deng et al. 2005).

We have expanded our discussion (page 8, line 755-766) to address the potential impact of such multivalent interactions between TRF2 and DNA, as well as the possible contribution of the TRFH domain in the gradual transition from a less stable to a more stable compact conformation.

}

Comment #6:

5) The authors show a gel with loss of binding when glutaraldehyde is added. Glutaraldehyde has been used for many years in electron microscopy and AFM to study many different DNA-protein complexes including nucleosomes (examples in references 3 and 4) and TRF2 (reference 5). Originally, the use of Glutaraldehyde was introduced and validated for electron microscopy. Indeed, it was shown that the structural and physical properties of nucleosomes were not perturbed by this crosslinking (references 6, 7 and references therein). That led to the use of this compound for studying different looped structures (references 8 to 10) and in AFM in the studies of chromatin (references 11 to 13 for examples) and later on for TRF2 itself in both techniques (references 5 and 14). In all these experiments, TRF2 experiments included, no report of any modified binding has been made. Moreover, some key AFM experiments showing the wrapping (DREEM) were performed without glutaraldehyde and after 20 minutes of incubation. These latter experiments show without doubt the wrapping since one can see the DNA around the protein (reference 2), again underlining the fact that experimental conditions in the present manuscript are unsuitable for studying TRF2 wrapping/chirality preference.

Response #6:

{

We would like to express our gratitude to the reviewer for the insightful comments and the recommended references.

Upon examination of the data obtained from our glutaraldehyde-gel assay, we have identified a crucial difference in experimental conditions between our study and previous AFM/EM/DREEM imaging analyses. In our gel assay, we incubated 1% glutaraldehyde with TRF2 first, and subsequently conducted the gel assay without removing any free glutaraldehyde present in the solution. This deviates from the earlier imaging analyses, where free glutaraldehyde molecules were eliminated.

As a result, our experimental conditions do not align with those of the previous imaging studies. Therefore, we have decided to exclude the data and associated text related to our gel experiment from our report.

}

Reviewer #3 (Remarks to the Author):

I read with interest the work Zhao et al entitled “TRF2 shows no detectable preference for chirality of supercoiled DNA and no 2 wrapping of telomeric DNA in single-DNA manipulation studies”.

This work focusses on DNA TRF2 interaction at the single molecule level with a mechanical approach using on magnetic tweezers. The authors measure interaction on dsDNA in a linear relaxed form to detect its effect on such a DNA, in a hairpin form to measure the affinity of the wild type and of variants on its target sequence or on a different DNA sequence, and more importantly on a supercoiled DNA as an asymmetric binding of TRF2 was published in a previous study.

The paper tests 3 points on 3 types of DNA substrates.

Comment #7:

Starting from the linear relaxed DNA to detect DNA extension modification by TRF2:

-There is no comment on the size of the shortening. 50nm is quite large. It could be compared to the size of the binding region ($25 \times 6 \times 0.34 = 51 \text{nm}$! that could eventually be corrected by an extension factor related to a WLC to consider the force) or to a footprint size of the enzyme. The proximity of the 2 distances (step size and size of the target sequence) brings confusion. A longer/shorter size of repeats would give a different value? This is to understand if the shortening measured is structural or related to the size of the target DNA that would in a way "collapse".

As mentioned in the text there could be interaction between different bound TRF2. Why not using a DNA construct with single or few spaced TTAGGG as, demonstrated in the “hairpin” section the affinity for this telomeric sequence is much higher than on random DNA? It would be important in view of the arguments developed about the different step sizes in the histogram (lines about 211). In addition with twice the minimum step size the PNA/DNA handles should be involved in the deformation. It is not a heavy work and it should clarify points on the detected states.

Response #7:

{

We are grateful for the reviewer's insightful comments regarding the step size data obtained from stretching linear DNA containing 25 repeats of TTAGGG. The reviewer accurately pointed out that the step size corresponds to a complete collapse of the linear DNA region, and the reviewer suspected that the step sizes may depend on the DNA length.

However, both our existing and new data, collected from much longer linear DNA strands (~7kb) with non-specific sequences, also display stepwise fluctuations and a step size distribution centered around ~25 nm (Fig. S4d). Notably, this step size is significantly smaller than the step sizes peaking at around 60 nm observed in 200 bp DNA containing 25 repeats of 5'TTAGGG (Fig. 1C) or 200 bp of random sequence

(Fig. S3). This result indicates that 1) The length of the DNA template, even when significantly longer, does not lead to a substantial increase in step size. 2) Instead, the average step size from the long DNA is smaller.

The smaller stepsize on long DNA may seem counterintuitive but can be explained by tension-induced cooperative clustering of DNA bending/wrapping proteins on the DNA strands. These proteins are positioned close to each other with the appropriate orientation, resulting in an overall extension of the DNA along the force direction to minimize free energy. Consequently, the reduction in average extension per nucleoprotein unit is smaller compared to shorter DNA strands that do not allow for such clustering. For a more detailed understanding of the physical mechanism, you can refer to a previous study conducted by the John Marko lab (Zhang and Marko 2010). A schematic figure from that study is provided below to illustrate this concept.

The existence of multiple compact binding modes of TRF2 and the large DNA wrapping step sizes of ~ 60 nm observed on 200 bp DNA under tension contradicts earlier findings from AFM imaging, which reported the formation of uniform nucleoprotein units on telomeric DNA, leading to consistent length reductions of approximately 27 nm per unit (Benarroch-Popivker et al. 2016). The data are more consistent with a subsequent study by the same research group that unveiled a more diverse range of nucleoprotein complexes on telomeric DNA (Kaur et al. 2016). These complexes can be broadly categorized into two groups: a small unit where DNA traverses TRF2 with slight bending, and a larger unit where DNA appears to wrap around the edges of TRF2 proteins (Kaur et al. 2016). In our study, the significant DNA compaction step sizes observed on the 200 bp DNA fragment containing 25 repeats of the 5'-TTAGGG sequence (as shown in Fig. 1C) align more closely with the larger nucleoprotein units reported in the latter study. It's worth noting that since 25 repeats of telomeric sequences may interact with more than one TRF2 molecule, it remains unclear whether the observed multiple binding modes represent distinct conformations of a single TRF2-DNA nucleoprotein complex or complexes involving multiple TRF2 molecules.}

We clarified this point in the Discussion section on page 7, line 628-644.
}

Comment #8:

-The time scale for the fluctuations between the different states do not depend on the force? Usually such fluctuations and their variations with force can bring interesting information on the energy landscape. This was often used in this field. Nothing is used here. Is there any reason? It is the non equilibrium aspect appearing in the F-H curves that prevented to do that? The fig2 pushes to deepen this aspect.

Response #8:

{
Indeed, the fluctuation time scale is highly dependent on force. However, we did not analyze the force-dependent lifetimes because it does not align with the purpose of Figure 2, which is to demonstrate the existence of two compact conformations with varying levels of mechanical stability.
}

Comment #9:

-A new element brought by this study is the existence of a more stable compact complex after some time as demonstrated in fig2. How long the time mentioned (> 500s) compares with the time between the injection and the start of the F-H curves? Those F_H curves present an hysteresis but are the curves almost the same after and before the appearance of this “stable” and compact state?

Response #9:

{
Figure 2 depicts a constant force experiment, where the change in bead height was measured under constant forces. The time elapsed between the injection of the protein solution and the initiation of the experiments, such as the force-loading scans shown in Figure 1, was generally around one minute.

Regarding the specific recordings of the bead height fluctuation at constant forces in Figure 1C and Figure 2, they were typically taken after several rounds of force-loading scans that produced the bead height-force data in Figure 1B. The tethers were initially subjected to high forces (> 5 pN) to unfold the compact TRF2-DNA complex. Subsequently, the forces were reduced to sub-pN levels to observe the dynamics of the subsequent formation of compact conformations. We searched the forces in this range at which reversible stepwise bead height changes could be observed.

The procedure of the how the constant force measurements were done have been provided in the revised manuscript (page 3, 190-196).

}

Comment #10:

-When it appears, this stable state last forever at low force? How long did the authors wait (500s being already quite long) ? This state brings a question on all the timescales of the experiments as underlined at different point in the text. Also in hairpin part, a more stable conformation would take longer to be removed, is this observed? Nevertheless this more stable state is interesting and should be investigated in future studies.

Response #10:

{
At a sub-pN force, when the more stable compact conformation formed following the less stable, dynamic conformation, it remained in the compact conformation at the same force till the ends of the recording in all experiments we performed. The typical time scale from entering the stable compact conformation till end was hundreds to

more than 1000 seconds. This information has been included in the Results subsection “Multiple DNA binding modes of TRF2 are revealed by single-DNA manipulation” (page 3, line 275-284).

Interestingly, once the stable compact conformation, it could be made dynamic again when slightly higher forces were applied to destabilize it. The force that destabilizes a compact biomolecular structure is directly related to the thermal stability of the structure via the following equation:

$$\mu = k_B T \ln \left(\frac{p_1(F)}{p_2(F)} \right) + \Delta\Phi_{12}(F),$$

where μ is the energy stored in the compact conformation, $p_{1(F)}$ and $p_{2(F)}$ are the force-dependent probabilities of the compact (state “1”) or extended (state “2”) states of the TRF2-DNA complex. The last term $\Delta\Phi_{12}(F) = -\int_0^F (x_1(f) - x_2(f))df$ is the force-induced shift of the free energy difference between the two states, which can be calculated based on the force-extension curves of the two states. In our data presented on figure 2, at a constant sub-pN force, the ration $p_{1(F)}/p_{2(F)}$ increases when the more stable conformation appeared, indicating an increase folding energy of μ .

From this equation, the critical force F_c at which the states have equal probability is determined by

$$\mu = -\int_0^{F_c} (x_1(f) - x_2(f))df$$

Therefore, it is clear that the more stable compact conformation, once formed at a sub-pN force, requires a higher force to destabilize it.

}

Comment #11:

-Line 269 the three states should be more clearly defined. If I am correct they correspond to the extended DNA form, then 2 short extensions forms but one being more stable than the other? This paragraph lacks some clarity for me.

Response #11:

{

The reviewer’s understanding of three states is correct. We have clarified the three states in the revised manuscript on page 3, line 275-284..

}

Comment #12:

-At the end of this part I am confused by the remark of the salt increase that shuts down the detection of the interaction while TRF2 remains on the DNA. It should clearly be more commented (it seems to imply at least 2 contact points one suppressed in the presence of high salt concentration). It raises the question TRF2 binding on ssDNA as

this could affect the detection of the hairpin part if TRF2 could stay on the ssDNA. Do the authors have some elements on this point?

Response #12:

{

The reviewer raised two questions for two different types of experiment. The first question pertains to the observation of the disappearance of DNA wrapping activity in DNA-bound TRF2 with increasing salt concentration. To bind DNA, we performed a buffer exchange to remove free TRF2 and reduce the salt concentration. We interpret this observation as an indication of a shift in the balance towards the non-wrapping DNA binding mode of TRF2 under higher salt concentrations. We agree that it indicates at least two TRF2-DNA contact points with one being suppressed by high salt concentration. We have included a sentence to clarify this point on page 3, line 255-260.

The second question is whether TRF2 binds ssDNA and whether such binding could affect our experiments in the hairpin study. Firstly, we confirm that TRF2 indeed exhibits ssDNA binding activity, as shown in Fig. S 13B and also in Fig. R1D in response to reviewer 1. Secondly, we would like to emphasize that our hairpin structure was not flanked by ssDNA handles. Instead, it is spanned between a PNA/ssDNA duplex to prevent binding to branched DNAs. The PNA/ssDNA duplex shows no detectable binding signals from TRF2, as explained in detail in response to reviewer 1.

}

Comment #13:

Supercoiled DNA to detect a binding dependant on the sign of the supercoiling :
-I start with a minor point in figure 3 symbols should be clarified as they are in fig 1. It is currently in the text but it would be better in the legend of the figure. Adding the force as insets in the figure would be nice too for non-experts of magnetic tweezers.

Response #13:

{

Legend has been added.

}

Comment #14:

-It is not clear whether the DNA template has the telomeric sequence. It is mentioned non-specific DNA so I would guess it is not the case but it should be the case for me, the specific binding to the telomeric sequence, and its chirality may depend on the sequence. The same experiment could be performed on a part of the cloned PUC57 used to prepare the linear construct.

Response #14:

{

We would like to clarify that the supercoiling data were acquired from a ~7-kb DNA fragment lacking telomere repeats. As suggested by the reviewer, we conducted

additional experiments using a 3-kb DNA fragment (PUC57) containing 25 repeats of TTAGGG. The obtained results from the DNA with the insert of telomeric repeats (Figure R 3 below, also included as Fig. S9 in the Supplementary Information) were found to be comparable to those obtained from the 7-kb DNA fragment without telomere repeats (Fig. S5 and S6). We have included a sentence to clarify this point on page 4, line 356-366.

This similarity suggests that the overall behavior of DNA supercoiling assays is not significantly influenced by the presence of a small segment of telomere repeats within a larger nonspecific DNA template. Important, no peak shift was observed for the DNA containing the telomere sequence, which suggests no significant chiral binding of TRF2 to the insert of the telomere sequence. We have added this information on page 4, line 403-409.

Figure R 3 TRF2 binding to supercoiled dsDNA containing 25 telomeric DNA repeats. A 3-kb DNA fragment (PUC57) containing 25 repeats of TTAGGG was wound/unwound to generate positive/negative supercoiled DNA at approximately 0.3 pN. In the presence of TRF2, the decrease in bead height indicates protein-induced compaction upon binding. The absence of a shift in the curve peak suggests no significant chiral binding of TRF2 to the inserted telomere sequence.

}

Comment #15:

-Indeed the proteins do not modify the curves presented in an asymmetric way. Nevertheless, at high force for $\sigma < 0$ the denatured or L-DNA disappears, this implies that the DNA buckles before denaturation at the same force. At what force it reappears? A force that prevents binding of TRF2? Could the authors comment?

Response #15:

{

The reviewer referred to the data presented in Figure 3D, which is reproduced below as Figure R 4. The figure illustrates that when a force of 1.3 pN is applied during the unwinding of DNA in the absence of TRF2 or with 5 nM TRF2, the formation of negative supercoiled DNA plectonemes is prohibited. Instead, a left-handed DNA structure or denatured DNA is formed. However, when the concentration of TRF2 is increased to 15 nM, the unwinding of the same DNA leads to the formation of supercoiled DNA plectonemes.

As pointed out by the reviewer, this suggests that TRF2 facilitates the buckling of DNA, enabling the formation of supercoiled plectonemes under higher forces. This observation supports the notion of TRF2 possessing DNA bending activity, as stated in our manuscript. We didn't test whether the TRF2 facilitated formation of the negative supercoiled plectonemes could occur at even higher forces.

Figure R 4 Refer back to the data presented in Figure 3D for a reprint of the corresponding figure.

}

Comment #16:

-Why not using the same TRF2 concentrations as in fig 1? In particular high concentrations to cover the DNA as much as possible?

Response #16:

{

Figure 1 utilized higher concentrations of TRF2. In the supercoiling assay, when concentrations exceeding 15 nM were employed, the compact DNA failed to relax into an extended conformation following the reversal of linking number density to $\sigma=0$, as depicted in the inserted diagram. This observation can be attributed to the presence of multiple DNA sites on TRF2, which potentially enhance the stabilization of plectonemes by bridging distant DNA sites when bound to multiple TRF2 molecules.

}

Comment #17:

-Finally on this part , I think an important measurement is missing: an extension vs time in the presence of supercoiling to detect the binding of individual TRF2 in a way similar to the work detailed in "Promoter unwinding and promoter clearance by RNA polymerase: Detection by single-molecule DNA nanomanipulation" by the group of T. Strick in PNAS. I think it was the first paper that used this approach but others have done since then. If any, differences in the binding/unbinding kinetics could bring differences in the affinity but also potential differences in the change of height induced by binding on the two sides could help deconvolute the bending and torsional components. These measurement would be more sensitive than the curves presented here I think. These would bring more complete proofs to a major claim of

the paper.

Response #17:

{

We would like to express our gratitude to the reviewer for the helpful suggestion on detecting whether binding induces a chiral change in DNA with high sensitivity. We conducted such measurements by introducing a (+) or (-) linking number change at a linking number density of ± 0.03 , while maintaining a fixed force of 0.3 pN to allow the formation of supercoiled plectonemes. We then added a solution of 20 nM TRF2 to observe any binding-induced changes in bead height (Figure R 5, also included in the Supplementary Information as Fig. S10). What we observed is that TRF2 induced further compaction of the DNA, regardless whether it was wound(+) or underwound(-). These findings indicate that TRF2 binding does not induce significant preferential chiral deformation of DNA, as there were no TRF2-induced significant relaxation of DNA supercoiling changes observed for both (+) and (-) supercoiled plectonemes. These results support our main conclusion. In this assay, to prevent any drag force from being applied to the DNA during the introduction of TRF2 solution, we utilized a PDMS membrane well-based perturbation isolation technology (Le et al. 2015).

Figure R 5 TRF2 binding on pre-formed supercoiling DNA. Representative time traces of TRF2 binding on positive supercoiling DNA (A-B) and negative supercoiling DNA (C-D). The shaded regions indicate the buffer exchange process for 20 nM TRF2.

}

Comment #18:

- The present conclusion on this part, in contradiction to previous results as noted in

the manuscript but with a discrepancy not justified, should be more robustly argued. Could the authors reproduce the experiment with TOP1 in bulk with their protein?

Response #18:

{

Due to the departure of our biochemistry collaborator, we are unable to repeat the bulk supercoiling assay. Instead, we conducted a single-molecule experiment to simulate the process of the bulk assay (Figure R 6, also included as Fig. S11 in the Supplementary Information). Initially, we held a DNA at zero linking number density at 0.3 pN and recorded the bead height. We then introduced 20 nM TRF2 with the perturbation isolation system and recorded the further reduction in the bead height caused by TRF2 binding. If TRF2 induces preferential chiral wrapping at its binding sites, then at TRF2-free regions, the DNA should form plectonemes with a opposite chirality to ensure unchanged linking number of the DNA.

With this background, if we introduce opposite winding to DNA, the plectonemes in the TRF2-free region should be relaxed (an effect similar to introducing TOP1), resulting in increased DNA extension. However, we did not observe any extension increases after the introduction of TOP1. FigureR8A shows the extension change of a DNA tether before and after introduction of 20 nM TRF2 at zero linking number change. DNA becomes shorter and exhibited dynamic fluctuation, indicating TRF2 mediated dynamic compaction of DNA. At time of 2000 s, the DNA was subjected to unwinding of -10 turns ($\Delta Lk = -10$), followed by winding of +20 turns ($\Delta Lk = +10$), and then unwinding of -10 turns ($\Delta Lk = 0$). As shown in the zoom-in in Figure R8B, the extensions at $\Delta Lk = +10$ and $\Delta Lk = -10$ are both not longer than the extension before the winding and unwinding processes. The results indicate that both winding and unwinding produced further plectonemes with the corresponding chirality, which are then stabilized by TRF2. Figure R8C-B show data repeated on a different DNA tether. Together, these observations are consistent with no strong chiral deformation of DNA by TRF2 binding.

Figure R 6 Detection chirality preference binding of TRF2 by winding/unwinding DNA (A, C) Representative time traces after adding 20 nM TRF2 and incubating for approximately 3000 seconds. Subsequently, the DNA was unwound and rewound (ΔLk shown in the bottom panels), and the change in DNA extension was examined before and after this process. (C, D) Zoomed-in time traces are displayed, corresponding to the red rectangles marked in panels A and C. }

Comment #19:

Hairpin part to measure the affinity of TRF2 to its specific site and the parts of the enzyme involved in binding:

- I think it would be worthwhile for non-specialists to explicitly indicate that there no force in the hairpin.

Response #19:

{
 We have clarified this important point using the following sentence: " At low forces ($F_b < F_c - \delta$), the stable DNA hairpin, which is excluded from force transmission, can interact with TRF2 in solution at zero force.", on page 5, line 462-464.
 }

Comment #20:

-The addition of non-specific (10 and 12 bases) DNA outside of the target site have no clear reason, for me, to be here in relation to the refolding of the hairpin. Can the authors give more elements? The authors use more than one telomeric sequence why

not using the same 25 repeats as in the linear part? In any case a longer DNA hairpin would avoid the remark made line 544. And the construct is not much more complex.

Response #20:

{

We chose not to use the 25-repeats of TTAGGG sequence for two reasons. Firstly, when we unfolded the hairpin with 25 TTAGGG repeats, it tended to fold into misfolded conformations, as indicated by different height levels after folding. This is mainly due to formation of various kinetically stable G4 structures before re-zipping occurs, as our buffer solution contains 150 mM KCl. Furthermore, with many repeats of TTAGGG, there is also a concern of hybridization with shifted registration of TTAGGG repeats and/or extensive G4 formation before re-zipping (as shown in Figure R 7).

Using 5 repeats of TTAGGG, the possibility of G4 formation before hybridization is drastically reduced. It also makes hybridization with shifted registration of TTAGGG repeats difficult as it will lead to formation of an unstable 6-bp dsDNA. The introduction of the non-specific (10 and 12 bases) DNA regions outside of the target site was to further ensure correct hybridization conformation of the DNA hairpin.

Secondly, our experiment was designed to probe the sequence-dependent binding that only depends on the C-terminal Myb-like domain and the energetic contribution from the N-terminus basic region. As previously reported (Court et al. 2005), five repeats of TTAGGG are sufficient for this purpose. It is worth noting that a longer TTAGGG segment would be necessary if the goal were to investigate the wrapping of DNA that involves the TRFH domain.

These reasons have been clarified on page 6, line 520-532.

Figure R 7 Mis-refolding of DNA hairpin containing 25 telomeric DNA repeats. (A) This schematic depicts the formation of a G-quadruplex on the G-rich strand of telomeric DNA, where the telomeric repeats undergo a shift and hybridize to an inappropriate location, leading to partial refolding of the DNA hairpin. (B) A representative time trace showing the change in bead height for a 200-bp hairpin that contains 25 repeats of TTAGGG. The hairpin undergoes sequential force jumps, ranging from 13.1 ± 1.3 pN to 15.3 ± 1.5 pN, and finally to 33.0 ± 3.3 pN, in the

absence of TRF2. The light orange shaded area indicates the correct position for hairpin refolding. However, during the second half of the time course, the hairpin did not refold into the correct position.

}

Comment #21:

-I have a technical question: why having a minimum of only 50 cycles, which is limited, and run bootstrap rather than using more statistics with longer acquisition times as the experiment seems to be self running and one cycle is not so long (with again the remark of an experiment where the protein could bind for more than 500s as mentioned in the linear DNA part)?

Response #21:

{

We ran 50 cycles on a single DNA, which took approximately 100 minutes. To prevent protein denaturation in the sample chamber, we replaced the TRF2 solution after the first 50 cycles. Additionally, to collect data from different DNA tethers, we repeated the 50-cycle process on a separate DNA tether.

}

Comment #22:

-Here again there are published approaches with a similar setup and configuration that could bring more quantitative information on a similar experiment. In "Single-molecule kinetic locking allows fluorescence-free quantification of protein/nucleic acid binding" Rieu et al. used a similar hairpin opening approach but that could bring kinetics elements in addition to the thermodynamic one as in the present paper. Why not using this approach? Does TRF2 precludes this? Timescales are too long? Nevertheless, to be more positive, the test on the effect of the DNA sequence and TRF2 domains binding properties is rather convincing if one excludes the possibility of a too short hairpin affecting the measure (see my previous remark).

Minor general remarks :

-Bibliography should be checked as it seems to be victim of a latex bad configuration (DNA in lowercase being the most typical). And there are few typos (in the supp mat also).

-Construct part in the protocol main text is surprisingly not so clear. A figure in the supp mat would help.

My general conclusion is that the work is reasonable, part of the conclusions (binding thermodynamics and partially the detection on linear DNA) are correctly justified but the supercoiling part, emphasized in the title, should be more supported and other parts require also complementary experiments. So the work should be improved to reach the point where a publication could be envisioned.

References:

- Benarroch-Popivker, D., S. Pisano, A. Mendez-Bermudez, L. Lototska, P. Kaur, S. Bauwens, N. Djerbi, C. M. Latrick, V. Fraasier, B. Pei, A. Gay, E. Jaune, K. Foucher, J. Cherfils-Vicini, E. Aeby, S. Miron, A. Londono-Vallejo, J. Ye, M. H. Le Du, H. Wang, E. Gilson, and M. J. Giraud-Panis. 2016. 'TRF2-Mediated Control of Telomere DNA Topology as a Mechanism for Chromosome-End Protection', *Mol Cell*, 61: 274-86.
- Kaur, P., D. Wu, J. Lin, P. Countryman, K. C. Bradford, D. A. Erie, R. Riehn, P. L. Opresko, and H. Wang. 2016. 'Enhanced electrostatic force microscopy reveals higher-order DNA looping mediated by the telomeric protein TRF2', *Sci Rep*, 6: 20513.
- Zhang, Houyin, and John F. Marko. 2010. 'Intrinsic and force-generated cooperativity in a theory of DNA-bending proteins', *Physical Review E*, 82: 051906.
- Court, Robert, Lynda Chapman, Louise Fairall, and Daniela Rhodes. 2005. 'How the human telomeric proteins TRF1 and TRF2 recognize telomeric DNA: a view from high-resolution crystal structures', *EMBO reports*, 6: 39-45.
- Deng, Yunxin, Artem K. Efremov, and Jie Yan. 2022. 'Modulating binding affinity, specificity, and configurations by multivalent interactions', *Biophysical Journal*, 121: 1868-80.
- Le, Shimin, Mingxi Yao, Jin Chen, Artem K. Efremov, Sara Azimi, and Jie Yan. 2015. 'Disturbance-free rapid solution exchange for magnetic tweezers single-molecule studies', *Nucleic Acids Research*, 43: e113-e13.

List of changes

1. In response to **comment #1** from reviewer 1. We have provided the following sentence in the second paragraph in Results, "The PNA/DNA hybrid handles were introduced used to suppress non-specific binding of TRF2 to the handles or to the fork of the hairpin, which were proven to be highly effective" and linked it to SI Fig. S1 and S13 for experimental evidence.
2. In response to **comment #2** from reviewer 2.

Main text change. The following sentences are added in the Discussion section (page 7, lines 664-675, "However, it does not rule out the possibility that TRF2 might have a weak preference for torsion-relaxed positively supercoiled DNA. There is a possibility that our assay might not be sensitive enough to detect if the induced chirality is not strong enough. We also note that the DNA in this single-molecule assay is under sub-pN forces, which is different from previous bulk assay utilizing circular DNAs that are not under similar mechanical constraint. This variation in experimental conditions could account for the previously observed generation of positive supercoils on circular DNA when TOPO I is present (21)."

Supplementary Information change: A new supercoiling assay with longer incubation time has been included in Supplementary Information: Time dependent of TRF2 binding on supercoiled DNA and Fig. S8, and referred to in the main text in the subsection "TRF2 binding does not have strong supercoiling chirality preference" on page 4, line 398-403.

3. In response to **comment #4** from reviewer 2, we have removed the sentences pertaining to DNA binding by the TRFH domain.
4. In response to **comment #5** from reviewer 2, a new paragraph on page 8, line 755-766 has been added to address the potential impact of such multivalent interactions between TRF2 and DNA, as well as the possible contribution of the TRFH domain in the gradual transition from a less stable to a more stable compact conformation.
5. In response to **comment #6** from reviewer 2, we have excluded the data and associated text related to our gel experiment from our report.
6. In response to **comment #7** from reviewer 3.

Main text change: Several sentences are added to the Discussion section (page 7, lines 628-644) discussing why the large step sizes (~50 nm) observed in Fig. 1C should be considered as a single step compaction by a TRF2 nucleoprotein complex instead of a collapse of a large DNA by multiple TRF2 nucleoprotein complexes. This is supported with a new Supplementary Information figure (Fig. S4D) showing regular folding and unfold steps on a much longer DNA.

Supplementary information change. A new supplementary figure (Fig. S4D) is included showing regular folding and unfold steps on a much longer DNA.

7. In response to **comment #9** from reviewer 3, the following sentences (page 3, 190-196) has been add to clarify the experimental protocol for data presented in Figure 2. "The tethers were initially subjected to high forces (> 5 pN) to unfold the compact TRF2-DNA complex. Subsequently, the forces were reduced to sub-pN levels to observe the dynamics of TRF2-mediated DNA conformation changes. We searched the forces at which reversible stepwise changes in the bead height could be observed."
8. In response to **comment #10** from reviewer 3, the following sentences (page 3, line 275-284) are added to provide the information on the typical time scale from entering the stable compact conformation till end the data recording. "At a sub-pN force, when the more stable compact conformation formed following the less stable, dynamic conformation, it remained in the compact conformation at the same force till the ends of the recording in all experiments we performed. The typical time scale from entering the stable compact conformation till end was hundreds to more than 1000 seconds."
9. In response to **comment #11** from reviewer 3, a sentence (page 3, line 275-284) "Three states of DNA conformation are revealed in these time traces, an extended DNA conformation, and two compact DNA conformations but one being more stable than the other." is added to clarify the DNA conformational states.
10. In response to **comment #12** from reviewer 3.

Main text change. A sentence (page 3, line 255-260) is added to discuss DNA binding mode of TRF2 under higher salt concentrations: “Furthermore, the observation that increasing the salt concentration can shift the DNA from a compact to an extended conformation without dissociation suggests that the TRF2-DNA complex involves at least two DNA contact points, and that the compaction caused by TRF2 is likely a result of an electrostatic interaction between TRF2 and the DNA.”

Supplementary Information change. New experimental evidence in Supplementary Information “PNA/ssDNA effectively suppresses TRF2’s branch-binding” is added to address the concern on whether TRF2 binds ssDNA and whether such binding could affect our experiments in the DNA hairpin based TRF2 binding affinity assay. Specifically, Fig. S13B is added to demonstrate the ssDNA binding activity of TRF2 that leads to ssDNA compaction, fork-binding of TRF2 that suppresses unzipping of DNA hairpin by force, and the elimination of ssDNA binding and fork binding activities by PNA clocking of the ssDNA handles (Fig. S13C-D).

11. In response to **comment #14** from reviewer 3, on page 4, line 356-366, we have incorporated new data into the Supplementary Information (Fig. S5), and added sentences of “We also tested whether chiral binding of TRF2 could take a longer time scale to occur, by conducting supercoiling experiments over different time scales up to 600 s. The resulting data still did not reveal a detectable chiral preference of DNA binding by TRF2.”

Page 4, line 403-409: The following sentences are added “A similar observation was made when using a 2.8 kb PUC-19 plasmid DNA with an insert of 25 repeats of 5'-TTAGGG (refer to Fig. S9). This suggests that TRF2 does not show any detectable preference for DNA supercoiling chirality in relation to the telomere sequence at this particular length (refer to Fig. S9).”

12. In response to **comment #17 and #18** from reviewer 3, we provided new data in Supplementary Information: Alternative method 1 to test chirality preference of TRF2 binding to DNA as Fig. S10 and Supplementary Information: Alternative method 2 to test chirality preference of TRF2 binding to DNA as Fig. S11. These information was updated in a new paragraph on page 5, line 410-431.
13. In response to **comment #19** from reviewer 3, we added the sentence of “ At low forces ($F_b < F_c - \delta$), the stable DNA hairpin, which is excluded from force transmission, can interact with TRF2 in solution at zero force.” on page 5, line 462-464.
14. In response to **comment #20** from reviewer 3, the sentences of “We found that using 5 repeats of TTAGGG, the possibility of G4 formation before hybridization is

negligible, and it produces sufficient length for specific binding mediated by the Myc domain. The non-specific dsDNA regions were added to guarantee full re-zipping of the hairpin at F_b (Supplementary Information 5: DNA construct for building DNA hairpin).” is add on page 6, line 520-532.

Figure changes in main text:

In response to **comment #13** from Reviewer 3, the text of applied force was add in Figure 3B-D. For better readability, all Naked DNA data was changed to black colour. In Figure 3A, square, circle and triangle symbols are used represents data recorded at 0.3 pN, 0.6 pN and 1.3pN, respectively.

Changes in Supplementary Information and figures: Mentioned in the responses 1-14.

New references:

1. In response to **comment #4** from Reviewer 2 and **comment #20** from Reviewer 3, a new reference describing a minimum binding site for TRF2 is added:
[48] R. Court, L. Chapman, L. Fairall, and D. Rhodes, How the human telomeric proteins trf1 and trf2 recognize telomeric dna: a view from high-resolution crystal structures, *EMBO reports* 6, 39 (2005).
2. In response to **comment #5** from Reviewer 2, a new reference describing the multivalent interactions is added:
[55] Y. Deng, A. K. Efremov, and J. Yan, Modulating binding affinity, specificity, and configurations by multivalent interactions, *Biophysical Journal* 121, 1868 (2022).
3. In response to **comment #12** from Reviewer 3, new references showing experimental evidences of protein concentration dependent dissociation are added:
[27] D. Skoko, B. Wong, R. C. Johnson, and J. F. Marko, Micromechanical analysis of the binding of dna-bending proteins hmgb1, nhp6a, and hu reveals their ability to form highly stable dna-protein complexes, *Biochemistry* 43, 13867 (2004).
[28] J. S. Graham, R. C. Johnson, and J. F. Marko, Concentration-dependent exchange accelerates turnover of proteins bound to double-stranded DNA, *Nucleic Acids Research* 39, 2249 (2010).
[29] C. E. Sing, M. Olvera de la Cruz, and J. F. Marko, Multiple-binding-site mechanism explains concentration-dependent unbinding rates of DNA-binding proteins, *Nucleic Acids Research* 42, 3783 (2014).
[30] R. I. Kamar, E. J. Banigan, A. Erbas, R. D. Giuntoli, M. O. de la Cruz, R. C. Johnson, and J. F. Marko, Facilitated dissociation of transcription factors from single dna binding sites, *Proceedings of the National Academy of Sciences* 114.
4. In response to **comment #17** from Reviewer 3, a new reference is added:
[40] A. Revyakin, R. H. Ebright, and T. R. Strick, Promoter unwinding and promoter clearance by rna polymerase: Detection by single-molecule dna nanomanipulation, *Proceedings of the National Academy of Sciences* 101, 4776 (2004).

Affiliations:

1. A new affiliation “Umeå university, KBC-huset (KB), Linnaeus väg 10, 901 87 Umeå, Sweden” is added for the co-corresponding author Sara Sandin.
2. A new affiliation of Joint School of National University of Singapore and Tianjin University, International Campus of Tianjin University, Binhai New City, Fuzhou, 350207, China is added for the corresponding author Jie Yan.

Reviewers' comments:

Reviewer #2 (Remarks to the Author):

Comment #2

Again, we disagree with the main point of the article stating that TRF2 has no chiral preference in binding. We do not argue about the results obtained but we do not understand why the authors keep negating the published results by giving such a misleading title to their article ("TRF2 shows no detectable preference for chirality of supercoiled DNA and no wrapping of telomeric DNA in single-DNA manipulation studies").

This title assumes that in any condition that may be chosen, chirality preference and wrapping cannot be detected. This is untrue.

The authors even concede that their conditions could preclude the observation of these events.

"While we did not observe DNA supercoiling by TRF2 in these assays, these data do not rule out the possibility that TRF2 might have a weak preference for torsion-relaxed positively supercoiled DNA. There is a possibility that our assay might not be sensitive enough to detect if the induced chirality is not strong enough. We also note that the DNA in this single-molecule assay is under sub-pN forces, which is different from previous bulk assay utilizing circular DNAs that are not under similar mechanical constraint. This variation in experimental conditions could account for the previously observed generation of positive supercoils on circular DNA when TOPO I is present." Thus, they admit that these events can exist but in conditions where forces are not applied. Scientific rigor and recognition of colleagues' work require thus require the title to be changed.

Comment #3

In our view, the hysteresis is more pronounced for positively supercoiled DNA when the binding of TRF2 is higher. The authors should calculate the skew factor of their traces and comment appropriately.

Comment #5

It is very disturbing that the authors negate the wrapping and chirality preference while admitting that the main provider of this (the TRFH domain) is a weak binder, probably involved in the establishment of a more stable complex formed with a lower kinetics that their experiments can achieve and most probably perturbed by the force they apply. Even more disturbing is the fact that the authors concede that the aforementioned force could preclude the observation of the very event they negate.

Reviewer #3 (Remarks to the Author):

The authors did quite a large amount of work compared to the first version of the paper. The arguments in the rebuttal letter, correctly reported in the manuscript in a concise way, and the new data (that represent some work indeed) are convincing enough for me to justify a publication as the manuscript brings enough information and sounds scientifically correct as far as one can judge from the manuscript. Some data are not in agreement with previously published ones but they nevertheless seem to be obtained in a reasonable way with the technique used here.

I have very minor comments.

I did not find what is the difference between fig S9 A and B. In fig S8 the red points are not described in the legend or in the text. It would help the reader.

Concerning the protocols I am still confused. In the main text the description of the hairpin is sent to the supp mat (line 839) but in the sup mat it looks like a description of the figure IA just as in the main text. Unless the sequence can fold as a hairpin but then colouring the complementary sequences would help. And there is no element about the constructs for the supercoiled DNA.

We thank the reviewers for the comments that have helped us to revise and improve the manuscript. All the comments from the reviewers have been addressed. Please find below the point-to-point replies to the comments. The list of main changes is added after the point-to-point replies. The changes in the main text are marked in **red**.

Responses to reviewers

Reviewer #2 (Remarks to the Author):

Comment #1

Again, we disagree with the main point of the article stating that TRF2 has no chiral preference in binding. We do not argue about the results obtained but we do not understand why the authors keep negating the published results by giving such a misleading title to their article ("TRF2 shows no detectable preference for chirality of supercoiled DNA and no wrapping of telomeric DNA in single-DNA manipulation studies"). This title assumes that in any condition that may be chosen, chirality preference and wrapping cannot be detected. This is untrue.

The authors even concede that their conditions could preclude the observation of these events.

"While we did not observe DNA supercoiling by TRF2 in these assays, these data do not rule out the possibility that TRF2 might have a weak preference for torsion-relaxed positively supercoiled DNA. There is a possibility that our assay might not be sensitive enough to detect if the induced chirality is not strong enough. We also note that the DNA in this single-molecule assay is under sub-pN forces, which is different from previous bulk assay utilizing circular DNAs that are not under similar mechanical constraint. This variation in experimental conditions could account for the previously observed generation of positive supercoils on circular DNA when TOPO I is present."

Thus, they admit that these events can exist but in conditions where forces are not applied. Scientific rigor and recognition of colleagues' work require thus require the title to be changed.

Response #1:

{

We thank again for the reviewer's critical assessment of the revised manuscript and glad to hear that he or she is not concerned about the results obtained. The main remaining criticism is about our claim that TRF2 has no chiral preference in binding.

We agree with reviewer's opinions, and have made the following revisions accordingly.

1. The title has been changed to 'Exploring TRF2-Dependent DNA Distortion Through Single-DNA Manipulation Studies,' which no longer implies that TRF2 does not induce chiral deformation under typical experimental conditions.

2. In the abstract, the words 'when DNA is under low tension' have been added to the sentence '...and TRF2's binding does not strongly favor a specific DNA supercoiling chirality,' to emphasize that our observation is specific to the experimental condition in our single-molecule assay where DNA is under low tension.

}

Comment #2:

In our view, the hysteresis is more pronounced for positively supercoiled DNA when the binding of TRF2 is higher. The authors should calculate the skew factor of their traces and comment appropriately.

Response #2:

{

We agree with the comment and have added the following sentences on page 4, lines 362-374: 'While these data do not clearly indicate a significant preference of TRF2 for a specific DNA supercoiling chirality in this assay, they do not definitively rule out a stronger right-handed DNA chiral wrapping model of the TRF2 nucleoprotein complex in the absence of tension. Indeed, despite the nearly symmetrical profile of the data obtained at low forces of 0.3 pN and 0.6 pN, the supercoiling curves appear slightly skewed toward the positive supercoiling density side at concentrations greater than 10 nM (Figure 3 B & C). This observation may suggest a subtle chiral deformation of DNA under low tension, which could become more pronounced if the force is further reduced.'

We have also revised the figure 3 B-C and figure S6 to indicate the subtle shift in the supercoiling center under these conditions.

}

Comment #3:

It is very disturbing that the authors negate the wrapping and chirality preference while admitting that the main provider of this (the TRFH domain) is a weak binder, probably involved in the establishment of a more stable complex formed with a lower kinetics that their experiments can achieve and most probably perturbed by the force they apply. Even more disturbing is the fact that the authors concede that the aforementioned force could preclude the observation of the very event they negate.

Response #3:

{

As mentioned in responses 1-2, we have clarified that the insignificant chiral deformation of DNA caused by TRF2 is limited to the specific condition in our single-molecule assay where the DNA is under certain tension. In response 2, we have also acknowledged that at lower tension and sufficiently high TRF2 concentration, the slight skew to the positive supercoiling density might indicate stronger chiral deformation of DNA by TRF2 if force is zero.

In addition, we have emphasized that our observation does not contradict with the observation from the previous supercoiling bulk assay in the 4th paragraph in Discussions, which is copied and pasted below.

`This result suggests that TRF2 does not exhibit a high affinity for positively supercoiled DNA when the DNA is subjected to sub-pN tension. However, this result does not contradict previous bulk biochemical assays, which demonstrated that TRF2 can induce positive supercoiling on circular DNA in the presence of TOPO I (Amiard et al., *Nat Struct Mol Biol*, **14**: 147-54). In those experiments, DNA molecules were not subjected to the same mechanical constraints as in our study. It is possible that our assay may not be sensitive enough to detect induced chirality if it is not sufficiently strong under the applied tension. Indeed, the slight skew of the supercoiling center toward the side with higher positive supercoiling density at low forces (<0.6 pN) and with TRF2 concentrations exceeding 10 nM suggests the potential for a more pronounced chiral deformation of DNA by TRF2 as the tension approaches zero. These variations in experimental conditions may account for the previously observed generation of positive supercoils on circular DNA in the presence of TOPO I (Amiard et al., *Nat Struct Mol Biol*, **14**: 147-54).`

}

Reviewer #3 (Remarks to the Author):

Comment #4:

The authors did quite a large amount of work compared to the first version of the paper. The arguments in the rebuttal letter, correctly reported in the manuscript in a concise way, and the new data (that represent some work indeed) are convincing enough for me to justify a publication as the manuscript brings enough information and sounds scientifically correct as far as one can judge from the manuscript. Some data are not in agreement with previously published ones but they nevertheless seem to be obtained in a reasonable way with the technique used here.

Response #4:

{

We appreciate the comment from Reviewer 3 that the new data are convincing enough to justify a publication and that the manuscript brings enough information and sounds scientifically correct as far as one can judge from the manuscript.

}

Comment #5:

I have very minor comments.

I did not find what is the difference between fig S9 A and B. In fig S8 the red points are not described in the legend or in the text. It would help the reader.

Response #5:

{

Figure S9 A and B present two representative datasets collected from two independent experiments illustrating TRF2 binding on supercoiled dsDNA containing 25 telomeric DNA repeats. This clarification is now provided in the revised R2 supplementary information.

In Figure S8, data points marked in red represent a 60-second time trace of bead height immediately after twisting the DNA from $\Delta Lk = +20$ or $\Delta Lk = -20$ back to $\Delta Lk = 0$. The mean value of the bead height during this time window is summarized in Figure

S8 C. This information has been incorporated into the revised R2 supplementary information.

}

Comment #6:

Concerning the protocols I am still confused. In the main text the description of the hairpin is sent to the supp mat (line 839) but in the sup mat it looks like a description of the figure 1A just as in the main text. Unless the sequence can fold as a hairpin but then colouring the complementary sequences would help. And there is no element about the constructs for the supercoiled DNA.

Response #6:

{

The reviewer is correct; the DNA sequence is designed to fold into a hairpin structure after one complementary strand is removed. Initially, the tethered DNA is double-stranded (dsDNA), with one strand tethered between the superparamagnetic bead and the coverslip surface. Subsequently, we apply a force exceeding the overstretching threshold under low-salt conditions to induce dissociation between the strands, resulting in a single-stranded DNA (ssDNA) tethered between the bead and the surface. This tethered ssDNA contains an inverted sequence that can form a hairpin structure when the force decreases. The remaining single-stranded DNA regions on either side of the hairpin are blocked using PNA.

A similar approach based on overstretching was employed to create a specific site under tensile force (as shown in Fig. 1A). In this scenario, the tethered single-stranded ssDNA, following the overstretching transition, lacks an inverted sequence, preventing the formation of a hairpin structure at low forces. Instead, it consists of 25 repeats of the sequence 5'-TTAGGG. To construct the specific telomere dsDNA site, a complementary strand containing 25 repeats of 5'-CCCTAA was introduced into the chamber. Following annealing, a dsDNA region with 25 repeats of 5'-TTAGGG was generated. In a similar manner, PNA was introduced to block the spanning ssDNA regions.

For a more detailed methodology, please refer to our previous publication (Zhao et al., *Curr Opin Chem Biol*, **53**: 106-17), where we have provided a comprehensive description of how the DNA is prepared and manipulated. In the methods section of this paper, we have briefly outlined the DNA preparation process and directed readers to our previous publication for further details

We have also added a new subsection 'Form DNA tethers' in MATERIALS AND METHODS to clarify how the single DNA tethers used in this study were formed.

}